# MULTI-GRANULARITY CORRESPONDENCE LEARNING FROM LONG-TERM NOISY VIDEOS

**Yijie Lin**[1]    **Jie Zhang**[2]    **Zhenyu Huang**[1]    **Jia Liu**[1]    **Zujie Wen**[3]    **Xi Peng**[1*]

[1]Sichuan University    [2]Beijing University of Posts and Telecommunications
[3]Dalian University of Technology
{linyijie.gm, pengx.gm}@gmail.com

## ABSTRACT

Existing video-language studies mainly focus on learning short video clips, leaving long-term temporal dependencies rarely explored due to over-high computational cost of modeling long videos. To address this issue, one feasible solution is learning the correspondence between video clips and captions, which however inevitably encounters the multi-granularity noisy correspondence (MNC) problem. To be specific, MNC refers to the clip-caption misalignment (coarse-grained) and frame-word misalignment (fine-grained), hindering temporal learning and video understanding. In this paper, we propose NOise Robust Temporal Optimal traNsport (Norton) that addresses MNC in a unified optimal transport (OT) framework. In brief, Norton employs video-paragraph and clip-caption contrastive losses to capture long-term dependencies based on OT. To address coarse-grained misalignment in video-paragraph contrast, Norton filters out the irrelevant clips and captions through an alignable prompt bucket and realigns asynchronous clip-caption pairs based on transport distance. To address the fine-grained misalignment, Norton incorporates a soft-maximum operator to identify crucial words and key frames. Additionally, Norton exploits the potential faulty negative samples in clip-caption contrast by rectifying the alignment target with OT assignment to ensure precise temporal modeling. Extensive experiments on video retrieval, videoQA, and action segmentation verify the effectiveness of our method. Code is available at https://lin-yijie.github.io/projects/Norton.

## 1 INTRODUCTION

Video-Language Pre-training (VLP) has emerged as a popular approach for video understanding (Miech et al., 2020; Bain et al., 2021; Ge et al., 2022; Wang et al., 2022c; Luo et al., 2020) in recent years. Although promising results have been achieved, the pioneer works are mainly devoted to learning short video clips while overlooking long-term temporal dependencies. In practice, it is generally acknowledged that the long-term temporal dependency plays an indispensable role in understanding the relationships and transitions over time in various applications such as video-paragraph retrieval (Yang et al., 2023b; Sun et al., 2022) and action segmentation (Tang et al., 2019).

To learn the long-term temporal correspondence from the long videos, one important challenge is the heavy demand for computation resources. For example, Han et al. (2022); Bertasius et al. (2021) employ long-form vision transformers to capture the temporal correlation, which involves computing cross-attention among every frame in long videos. As long videos are typically composed of a sequence of short video clips according to ASR timestamps (Miech et al., 2020), an alternative approach is to explore the temporal correlation among video clips and captions. For instance, TempCLR (Yang et al., 2023b) uses Dynamic Time Warping (Müller, 2007; Cuturi & Blondel, 2017; Zhou & Torre, 2009) to measure the sequential distance between video clips and captions, and incorporates the temporal correlation across clips by contrasting the video with the paragraph. This strategy is remarkably efficient than directly modeling the entire video, making it an attractive option for learning long-term temporal correspondence.

However, dividing long videos into short clips would inevitably introduce an accompanied challenge, *i.e.*, multi-granularity noisy correspondence (MNC). As shown in Fig. 1, MNC refers to the

---

*Corresponding author.

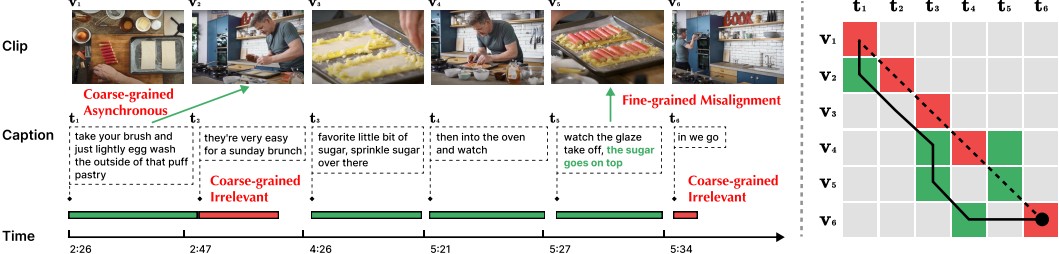

Figure 1: Our observation on multi-granularity noisy correspondence (MNC) in video understanding. (*Left*) The green timeline denotes the alignable captions while the red timeline indicates the unalignable captions. The green text in $\mathbf{t}_5$ denotes partially correlated words w.r.t $\mathbf{v}_5$. (*Right*) The dashed line represents the original alignment according to timestamps and the red block indicates the misaligned clip-caption pair. The green block denotes the ground-truth alignment. The solid line denotes the re-alignment by Dynamic Time Warping (Müller, 2007) which struggles to handle noisy correspondence well.

misaligned video-text pairs at two different granularities: i) *Coarse-grained misalignment (Clip-caption)*. Coarse-grained misalignment includes *asynchronous* and *irrelevant* misalignments according to whether a clip/caption is alignable with the captions/clips in the long video. To be specific, asynchronous misalignment refers to temporal misalignment between subtitles and visual clips, *e.g.*, $\mathbf{t}_1$ in Fig. 1. It often occurs when people explain their actions before or after actually performing them, resulting in the mismatch between the order of statements and actions. On the other hand, irrelevant misalignment refers to irrelevant or meaningless captions that cannot be aligned with any available video clips (*e.g.*, $\mathbf{t}_2$ and $\mathbf{t}_6$ in Fig. 1), and vice versa for video clips. According to Han et al. (2022), only 30% of clip-caption pairs are visually aligned in HowTo100M (Miech et al., 2019), with even fewer 15% being naturally well-aligned; ii) *Fine-grained misalignment (Frame-word)*. Within each video clip, the narration sentences may only partially correlate with the visual frames. As depicted in Fig. 1, "the sugar goes on top" in $\mathbf{t}_5$ is strongly correlated with visual content $\mathbf{v}_5$ while the action "watch the glaze take off" is uncorrelated. Irrelevant words or frames can distort the identification of crucial ones and result in inaccurate similarity measurements, further contaminating the clip-caption alignment. Note that only a few methods (Han et al., 2022) consider the coarse-grained misalignment problem in temporal learning while none of them realize this fine-grained misalignment problem. Undoubtedly, MNC poses a significant obstacle to effective temporal modeling.

To this end, we propose NOise Robust Temporal Optimal traNsport (Norton), a unified optimal transport approach for addressing multi-granularity noisy correspondence in temporal learning. Specifically, Norton proposes a video-paragraph and a clip-caption contrastive loss based on optimal transport (OT) to explore the temporal correlations.

In video-paragraph contrast, Norton employs OT to measure sequence distances between video clips and captions from a fine-to-coarse perspective. To handle fine-grained misalignment, Norton incorporates a token-wise soft-maximum operator to identify crucial words and key frames within each clip-caption pair. This operator improves the measurement of clip-caption similarity from fine-grained multi-modal interactions. Building upon this clip-caption similarity, Norton establishes a flexible assignment between clips and captions by maximizing the global alignment similarity of OT. Based on the transport assignment, Norton realigns each video clip to multiple related captions, and vice versa, thereby mitigating the asynchronous misalignment. To further address the irrelevant misalignment, Norton introduces an alignable prompt bucket which serves as a candidate alignable target for noisy clips or captions. By discarding the ones aligned to the bucket, Norton effectively filters out meaningless content during the OT process. Note that our late interaction between clips and captions through OT alleviates the computational cost of directly modeling long videos.

In clip-caption contrast, Norton tackles the faulty negative problem (Chuang et al., 2020; Yang et al., 2021b) through OT. Specifically, semantically similar clip and captions would be wrongly treated as negatives in contrastive learning (Chen et al., 2020; Lin et al., 2021; 2022; Liu et al., 2022a) and impact the clip-wise representation. Norton leverages OT assignments of within-batch clip-caption pairs as additional supervision in clip-caption contrastive loss, which exploits potential faulty negative samples and improves temporal learning.

The main contributions of this work are summarized below:

- We reveal multi-granularity noisy correspondence problem in temporal learning, which refers to coarse-grained asynchronous and irrelevant misalignments, as well as fine-grained misalignment.
- We achieve efficient and robust correspondence learning by incorporating several innovative components such as the soft-maximum operator, alignable prompt bucket, and faulty negative exploitation within the optimal transport framework. Extensive experiments on various tasks including video retrieval, videoQA, and action segmentation verify its effectiveness.

## 2 RELATED WORK

**Video Temporal Learning.** Temporal learning is a critical yet challenging topic in video understanding. Traditional works focus on integrating spatial-temporal operations into convolution (Feichtenhofer et al., 2019) or Transformer architectures (Bertasius et al., 2021; Wang et al., 2023; Sun et al., 2022). Inspired by image-language pre-training approaches (Radford et al., 2021; Jia et al., 2021), recent works leverage natural language to guide video temporal learning. Among these works, one scheme is "sorting the clips" (Zellers et al., 2021; Zeng et al., 2023a;b; Ma et al., 2023) which involves ranking the video clips according to their sequential sentences. While effective, this framework generally requires encoding long video into one sequence and entails significant computational resources. Another type of scheme proposes to leverage Dynamic Time Warping (Yang et al., 2023b; Müller, 2007; Dvornik et al., 2021) to measure the sequence distance between video clips and captions, and achieve temporal learning by aligning the video with the corresponding paragraph.

Although promising results have been achieved, existing temporal learning methods suffer from the noisy correspondence problem where the ground truth order of captions w.r.t. video clips does not conform to the original timestamp order. This issue can significantly impact temporal learning, leading to suboptimal results for sorting-based and DTW-based approaches. Different from these works, this paper is dedicated to solving noisy correspondence in temporal learning and accordingly proposes an MNC-robust optimal transport framework that effectively measures sequence similarity between noisy video and paragraph.

**Noisy Correspondence Learning in Video-language Pre-training.** Video-language pre-training has achieved promising progress thanks to large-scale datasets such as HowTo100M (Miech et al., 2019). As the text description is often not well-aligned to the visual content (Han et al., 2022), noisy correspondence learning (Huang et al., 2021; Gao et al., 2021) becomes a new fashion in VLP. To be specific, MIL-NCE (Miech et al., 2020) first studies this problem by simply aligning each video clip with multiple adjacent sentences to mitigate the impact of noise. TAN (Han et al., 2022) proposes a co-training strategy that uses mutual agreement to filter out the noisy pairs. Different from the above on-the-fly noise rectified methods, Decembert (Tang et al., 2021) generates high-quality video descriptions using an off-the-shelf image captioning model from a data collection aspect.

Our method differs from existing works in two key aspects. First, the above noisy correspondence methods only consider coarse-grained asynchrony while ignoring the frame-word misalignment problem. In contrast, we point out that fine-grained misalignment can impact temporal learning and accordingly propose a unified optimal transport approach that effectively addresses noisy correspondence at both coarse and fine-grained levels. Second, our method is computationally efficient with a low memory cost. It operates in a bootstrapping manner without requiring additional models, *e.g.*, dual networks (Han et al., 2022), momentum networks (Li et al., 2021; Han et al., 2022), or image caption models (Tang et al., 2021). These advantages make our approach more practical and scalable for real-world applications.

**Optimal Transport.** OT is originally proposed to depict the distance between two probability distributions. Recently, OT has gained significant attention in various fields such as domain adaptation (Xu et al., 2020), clustering (Caron et al., 2020), document matching (Yu et al., 2022; Kusner et al., 2015), and sequence alignment (Su & Hua, 2017; Liu et al., 2022b). However, none of these works specifically focus on the alignment of video and text, which is the primary focus of our research. In addition to addressing the traditional sequence alignment, we point out the fine-grained misalignment problem that is specific to video-text learning. Experimental results show that the proposed multi-grained alignment effectively improves temporal learning.

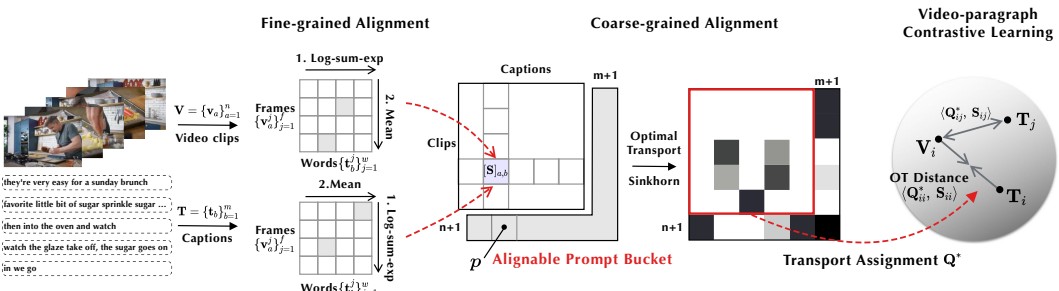

Figure 2: Overview of our multi-granularity correspondence learning. We perform video-paragraph contrastive learning to capture long-term temporal correlations from a fine-to-coarse perspective. Specifically, we first utilize the log-sum-exp operator on the frame-word similarity matrix to obtain fine-grained similarity between clip and caption. Additionally, we append an alignable prompt bucket on the clip-caption similarity matrix to filter out the irrelevant clips or captions. By applying Sinkhorn iterations on the clip-caption similarity matrix, we effectively tackle the asynchronous problem and obtain the optimal transport distance as the video-paragraph similarity.

## 3  METHOD

In this section, we first introduce the overall pre-training objective of Norton in Section 3.1. Subsequently, we elaborate on our multi-granularity correspondence learning in Section 3.2 and explain how to exploit the faulty negative samples in clip-caption contrastive learning in Section 3.3.

### 3.1  PRE-TRAINING OBJECTIVE

Given an instructional video dataset $\mathcal{D} = \{\mathbf{V}_i, \mathbf{T}_i\}_{i=1}^N$, where $\mathbf{V}_i$ and $\mathbf{T}_i$ represent the video and paragraph of $i$-th instance, we formulate each video/paragraph as a sequence of video clips/captions according to the ASR timestamps. Specifically, we mark the video clips and captions in $i$-th video as $\{\mathbf{v}_a\}_{a=1}^n$ and $\{\mathbf{t}_b\}_{b=1}^m$. Here $\{\mathbf{v}_a^j\}_{j=1}^f$ and $\{\mathbf{t}_b^j\}_{j=1}^w$ represent the frames and words within $\mathbf{v}_a$ and $\mathbf{t}_b$, where $f$ and $w$ represent the length of the clip and caption. Based on the above definitions, we propose the following training objectives:

$$\mathcal{L} = \mathcal{L}_{\text{clip}} + \lambda \mathcal{L}_{\text{video}}, \tag{1}$$

where video-paragraph contrastive loss $\mathcal{L}_{\text{video}}$ explores the temporal correlations between the long video $\mathbf{V}_i$ and its corresponding paragraph $\mathbf{T}_i$ through a novel noise robust temporal optimal transport distance. The clip-caption contrastive loss $\mathcal{L}_{\text{clip}}$ exploits potential faulty negative samples to improve clip representation and ensure accurate temporal modeling. We will elaborate on these two losses in the following sections.

### 3.2  CORRESPONDENCE LEARNING VIA ROBUST OPTIMAL TRANSPORT

As long videos are typically composed of a sequence of short video clips, we propose to use the optimal transport distance between video clips and captions as the similarity criterion for video-paragraph contrastive learning in a robust and efficient way.

Let $\mathbf{S} \in \mathbb{R}^{n \times m}$ denote the clip-caption similarity matrix where $[\mathbf{S}]_{a,b}$ measures the similarity between clip $\mathbf{v}_a$ and caption $\mathbf{t}_b$. $\mathbf{Q} \in \mathbb{R}_+^{n \times m}$ denotes the corresponding transport assignment where $[\mathbf{Q}]_{a,b}$ represents the probabilities of aligning $\mathbf{v}_a$ with $\mathbf{t}_b$. Optimal transport seeks to establish a flexible alignment between clips and captions by maximizing global similarity $\langle \mathbf{Q}, \mathbf{S} \rangle = \text{tr}(\mathbf{Q}^\top \mathbf{S})$. Formally, the objective of optimal transport is defined as follows:

$$\begin{aligned} \max_{\mathbf{Q} \in \mathcal{Q}} \quad & \langle \mathbf{Q}, \mathbf{S} \rangle + \varepsilon H(\mathbf{Q}) \\ \text{s.t.} \quad & \mathcal{Q} = \left\{ \mathbf{Q} \in \mathbb{R}_+^{n \times m} \mid \mathbf{Q}\mathbf{1}_m = \boldsymbol{\mu}, \mathbf{Q}^\top \mathbf{1}_n = \boldsymbol{\nu} \right\}. \end{aligned} \tag{2}$$

where $\mathbf{1}_m$ represents the vector of ones in dimension $m$, $\boldsymbol{\mu} \in \mathbb{R}^n$ and $\boldsymbol{\nu} \in \mathbb{R}^m$ indicate the relative importance of each clip or caption. Since each clip or caption is sampled independently, we choose uniform probability distribution $\boldsymbol{\mu} = \frac{1}{n}\mathbf{1}_n$ and $\boldsymbol{\nu} = \frac{1}{m}\mathbf{1}_m$ to assign equal weight to each

instance following Su & Hua (2017). $H(\mathbf{Q})$ is an entropy regularizer derived from the optimization perspective (Cuturi, 2013) and $\varepsilon$ controls its smoothness.

As illustrated in Eq. (2), optimal transport can realign each clip or caption to multiple related captions or clips based on global similarity, thus effectively resolving the potential asynchronous misalignment problem between the two modalities. The optimal $\mathbf{Q}^*$ of Eq. (2) has a simple normalized exponential matrix solution by Sinkhorn fixed point iterations (Cuturi, 2013),

$$\mathbf{Q}^* = \mathrm{Diag}(\boldsymbol{\kappa}_1) \exp\left(\mathbf{S}/\varepsilon\right) \mathrm{Diag}(\boldsymbol{\kappa}_2),$$

with iteratively updated $\boldsymbol{\kappa}_1 \leftarrow \boldsymbol{\mu}./\left(\exp\left(\mathbf{S}/\varepsilon\right)\boldsymbol{\kappa}_2\right),\ \boldsymbol{\kappa}_2 \leftarrow \boldsymbol{\nu}./\left(\exp\left(\mathbf{S}^\top/\varepsilon\right)\boldsymbol{\kappa}_1\right),$ (3)

where $\boldsymbol{\kappa}_1 \in \mathbb{R}^n$, $\boldsymbol{\kappa}_2 \in \mathbb{R}^m$ are the non-negative left and right scaling vectors. By utilizing OT distance between clips and captions as the video-paragraph similarity, our video-paragraph contrastive loss captures the long-term temporal dependencies as follows,

$$\mathcal{L}_{\text{video}} = -\sum_{i=1}^{N}\left(\log\frac{\exp\left(\langle\mathbf{Q}_{ii},\ \mathbf{S}_{ii}\rangle/\tau\right)}{\sum_{j=1}^{N}\exp\left(\langle\mathbf{Q}_{ij},\ \mathbf{S}_{ij}\rangle/\tau\right)} + \log\frac{\exp\left(\langle\mathbf{Q}_{ii},\ \mathbf{S}_{ii}\rangle/\tau\right)}{\sum_{j=1}^{N}\exp\left(\langle\mathbf{Q}_{ji},\ \mathbf{S}_{ji}\rangle/\tau\right)}\right),$$ (4)

where $\mathbf{S}_{ij} \in \mathbb{R}^{n\times m}$ is the clip-caption similarity matrix between video $\mathbf{V}_i$ and paragraph $\mathbf{T}_j$, $\mathbf{Q}_{ij}$ is the corresponding transport assignment of $\mathbf{S}_{ij}$, and $\tau$ is a learnable temperature initialized as 0.07. Note that when calculating Eq. (4), we stop the gradient of the transport assignment $\mathbf{Q}$ to keep the stability of our video-paragraph contrastive loss. To ensure the discriminative capacity of the model, we search the nearest videos as the hard negative samples following Xu et al. (2021). By using optimal transport to measure sequence distance instead of directly modeling the long videos, our method significantly reduces computational cost. A detailed training efficiency discussion is placed in Appendix C.

However, the optimal transport objective Eq. (2) still has some limitations: i) OT estimates the sequence distance based on clip-caption similarity (coarse-grained), leaving word-frame misalignment (fine-grained) problem unexplored; ii) OT requires each source instance must exactly map to the targets, which is not practical when dealing with a large amount of meaningless text. To address these challenges, we propose a soft-maximum operator for fine-grained alignment and an alignment prompt bucket to filter out meaningless clips and captions for noise robust distance estimation.

**Fine-grained Alignment.** Most previous works (Xu et al., 2021; Yang et al., 2023b; Han et al., 2022) typically encode frames or words to a global feature using [CLS] token or averaging the frame or word embeddings (*e.g.*, $\mathrm{AvgPool}(\{\mathbf{v}_a^j\}_{j=1}^{f})$). However, such strategies neglect fine-grained interactions between modalities and do not address the problem of frame-word misalignment.

To address this issue, we propose a cross-modal late interaction mechanism to identify crucial words and key frames for fine-grained alignment inspired by Yao et al. (2022); Wang et al. (2022b). Specifically, we define the fine-grained similarity between clip $\mathbf{v}_a$ and caption $\mathbf{t}_b$ as follows:

$$[\mathbf{S}]_{a,b} = \frac{1}{2}\left(\frac{1}{f}\sum_{i=1}^{f}\alpha\log\left(\sum_{j=1}^{w}\exp(\frac{\mathbf{v}_a^i\cdot\mathbf{t}_b^j}{\alpha})\right) + \frac{1}{w}\sum_{i=1}^{w}\alpha\log\left(\sum_{j=1}^{f}\exp(\frac{\mathbf{t}_b^i\cdot\mathbf{v}_a^j}{\alpha})\right)\right).$$ (5)

Take the front part for example, for each frame in the video clip, we identify the most important words through a soft-maximum operation, *i.e.*, log-sum-exp approximation (Beck & Teboulle, 2012), and then compute the average soft-maximum similarities of all frames as shown in Fig. 2. Similarly, for each textual token, we also find its related video frames in the latter part of Eq. (5). The parameter $\alpha$ magnifies the importance of the most relevant words or frames. As $\alpha$ approaches 0, the log-sum-exp approximates the maximum. Specifically, this soft-maximum operation allows us to reduce the negative influence of background words or frames on clip-caption similarity estimation.

Though inspired from Wang et al. (2022b); Yao et al. (2022), our method differs in several aspects. Firstly, we introduce a straightforward log-sum-exp operator as a soft approximation of the maximum. This allows us to concentrate on more crucial words, making it particularly well-suited for video content as opposed to images. Experiments in Table 7 demonstrate that our design yields a substantial improvement compared to solely focusing on the most important item. Secondly, we leverage the estimated clip-caption similarity for sequence alignment, effectively enhancing temporal learning. In contrast, Wang et al. (2022b) exclusively concentrates on clip-caption alignment.

**Alignable Prompt Bucket.** Optimal transport requires every source instance to exactly map to the targets. Yet, in real-world scenarios, a significant amount of captions and video clips might be noisy or irrelevant that cannot be aligned, *i.e.*, coarse-grained irrelevant misalignments. Motivated by Sarlin et al. (2020), we propose an innovative solution that uses an alignable prompt bucket (APB) to filter out semantic irrelevant clips and captions. As shown in Fig. 2, the prompt bucket consists of one new row and column, filled with the same value $p$. The prompt bucket is appended to the similarity matrix $\mathbf{S}$ that

$$[\bar{\mathbf{S}}]_{a,m+1} = [\bar{\mathbf{S}}]_{n+1,b} = [\bar{\mathbf{S}}]_{n+1,m+1} = p, \ [\bar{\mathbf{S}}]_{a,b} = [\mathbf{S}]_{a,b}, \ \forall a \in [1,n], \ b \in [1,m]. \quad (6)$$

When calculating the transport distance given $\bar{\mathbf{S}}$, each video clip can be aligned with either available captions or the prompt bucket. Substituting Eq. (2) with Eq. (6), we obtain the final optimal transport assignment by dropping the last row and column of the transport assignment, *i.e.*, $\bar{\mathbf{Q}}^* = \bar{\mathbf{Q}}^*_{1:n,1:m}$.

From an intuitional viewpoint, the prompt value $p$ in Eq. (6) serves as a similarity margin that distinguishes between alignable and unalignable clips and captions. If a video clip $\mathbf{v}_a$ lacks an alignable caption, its pairwise similarities with the set of captions $\{\mathbf{t}_b\}_{b=1}^m$ are generally small. Consequently, if the margin $p$ is larger than these pairwise similarity values, $\mathbf{v}_a$ is forced to align with the prompt bucket and subsequently filtered from the transport assignment. In our implementation, we determine the value of $p$ as the bottom 30% similarity of the original aligned clip-caption pairs in a data-driven manner.

### 3.3 Clip-caption Alignment via Faulty Negative Exploitation

Since self-supervised contrastive learning (He et al., 2020) relies on the random sampling of negative instances, captions that are semantically similar to the anchor clips can be treated as faulty negatives (Han et al., 2020; Zolfaghari et al., 2021), and vice versa. However, the existing one-hot target used in contrastive learning penalizes all negative predictions regardless of their correlations.

To mitigate this issue, we propose to exploit the faulty negatives through optimal transport. Let $\hat{\mathbf{S}} \in \mathbb{R}^{B \times B}$ denotes the within-batch clip-caption similarity matrix where $B$ represents the number of clips/captions for all videos in the batch. We apply optimal transport on the similarity matrix $\hat{\mathbf{S}}$,

$$\max_{\hat{\mathbf{Q}} \in \hat{\mathcal{Q}}} \ \langle \hat{\mathbf{Q}}, \ \hat{\mathbf{S}} \rangle + \varepsilon H(\hat{\mathbf{Q}}) \quad \text{s.t.} \ \hat{\mathcal{Q}} = \left\{ \hat{\mathbf{Q}} \in \mathbb{R}_+^{B \times B} \mid \hat{\mathbf{Q}} \mathbf{1}_B = \frac{1}{B} \mathbf{1}_B, \hat{\mathbf{Q}}^\top \mathbf{1}_B = \frac{1}{B} \mathbf{1}_B \right\}, \quad (7)$$

where the transport assignment $\hat{\mathbf{Q}}$ attempts to realign the clips with similar captions (*i.e.*, faulty negatives). After implementing the Sinkhorn algorithm described in Eq. (3), we utilize the clip-wise realigned targets $\hat{\mathbf{Q}}^*$ as additional supervision for contrastive learning,

$$\mathcal{L}_{\text{clip}} = - \sum_{i=1}^{B} \sum_{j=1}^{B} [\mathbf{T}]_{i,j} \left( \log \frac{\exp([\hat{\mathbf{S}}]_{i,j}/\tau)}{\sum_{k=1}^{B} \exp([\hat{\mathbf{S}}]_{i,k}/\tau)} + \log \frac{\exp([\hat{\mathbf{S}}]_{i,j}/\tau)}{\sum_{k=1}^{B} \exp([\hat{\mathbf{S}}]_{k,j}/\tau)} \right), \mathbf{T} = (1 - \beta) \, \mathbf{I}_B + \beta \hat{\mathbf{Q}}^*,$$
$$(8)$$

where $\beta$ is a weighted parameter that balances the identity target $\mathbf{I}_B$ and realigned targets $\hat{\mathbf{Q}}^*$. By replacing identity matrix $\mathbf{I}_B$ with estimated soft-alignment probabilities, the model can recalibrate the attractive and repulsive forces between clips and captions. Specifically, the entire training batch is treated as a support set (Patrick et al., 2021) with a subset of relevant clips and captions. Our method enables the detection and correction of potential faulty negatives within the set.

## 4 Experiments

We verify the effectiveness of Norton in comprehending both long and short videos across a range of downstream tasks. Additionally, we perform extensive ablation studies to analyze the impact of different design choices on the model's performance. For comprehensive training details, training efficiency results, and additional experiments please refer to the Appendix.

### 4.1 Comparisons on Video-paragraph Retrieval

As the main contribution of this work lies in long-term temporal learning, we first evaluate our method on the video-paragraph retrieval task. The objective of this task is to accurately find the corresponding video using a set of sentence queries that describe different parts of the long video.

Table 1: Video-paragraph retrieval on YouCookII (*Background Removed*). The best and second-best results are **bold** and underlined, respectively.

| Approach | Measure | R@1 | R@5 | R@10 |
|---|---|---|---|---|
| MIL-NCE (Miech et al., 2020) | Cap. Avg. | 43.1 | 68.6 | 79.1 |
| HT100M (Miech et al., 2019) | Cap. Avg. | 46.6 | 74.3 | 83.7 |
| MCN (Chen et al., 2021) | Cap. Avg. | 53.4 | 75.0 | 81.4 |
| VideoCLIP (Xu et al., 2021) | Cap. Avg. | 74.5 | 94.5 | **97.9** |
| TempCLR (Yang et al., 2023b) | Cap. Avg. | 74.5 | 94.6 | 97.0 |
| Norton (Ours) | Cap. Avg. | **75.5** | **95.0** | 97.7 |
| VideoCLIP (Xu et al., 2021) | DTW | 56.0 | 89.9 | 96.3 |
| TempCLR (Yang et al., 2023b) | DTW | 83.5 | 97.2 | 99.3 |
| Norton (Ours) | DTW | **88.7** | **98.8** | **99.5** |
| VideoCLIP (Xu et al., 2021) | OTAM | 52.8 | 89.2 | 95.0 |
| TempCLR (Yang et al., 2023b) | OTAM | 84.9 | 97.9 | 99.3 |
| Norton (Ours) | OTAM | **88.9** | **98.4** | **99.5** |

Table 2: Video-paragraph retrieval on YouCookII (*Background Kept*).

| Approach | R@1 | R@5 | R@10 |
|---|---|---|---|
| Cap. Avg. | | | |
| VideoCLIP | 73.6 | **94.7** | **98.4** |
| TempCLR | 71.7 | 94.5 | 97.9 |
| Norton (Ours) | **74.8** | **94.7** | **98.4** |
| DTW | | | |
| VideoCLIP | 55.7 | 93.1 | **98.9** |
| TempCLR | 70.4 | 93.8 | 97.9 |
| Norton (Ours) | **76.1** | **95.0** | 97.7 |
| OTAM | | | |
| VideoCLIP | 56.6 | 92.8 | **98.9** |
| TempCLR | 72.2 | 94.5 | 97.7 |
| Norton (Ours) | **73.6** | **94.7** | 97.7 |

**Setup and Metric.** We evaluate the zero-shot performance of our method in two different settings, namely, *Background Removed* and *Background Kept*. The former setting discards the text-uncorrelated video clips based on the timestamps, while the latter uses the full video. As timestamps may not always be available, paragraph retrieval with background is a more realistic scenario. To provide a comprehensive evaluation, we employ three standard strategies, namely, Cap. Avg. (Caption Average), DTW, and OTAM (Ordered Temporal Alignment Module (Cao et al., 2020)). Specifically, Cap. Avg. matches one clip for each caption and retrieves the video with the most matched clips. DTW and OTAM calculate the sequence distance by accumulating the clip-caption distance based on chronological order. We report recall metrics R@1, R@5, and R@10 for all setups. Specifically, R@1 indicates how often the correct prediction is the first result, which is highly desirable in many applications, while R@10 provides a wider scope and may be less critical as users typically focus on the top few results in practical scenarios.

**Datasets.** We conduct the evaluation on YouCookII (Zhou et al., 2018) where the testing data consists of 436 videos with 3,350 clip-caption pairs in total. The videos existing in YouCookII have been removed from Howto100M (Miech et al., 2019) following the same protocol as previous works (Miech et al., 2020; Xu et al., 2021; Yang et al., 2023b).

**Results.** i) *Background Removed*: As shown in Table 1, TempCLR (Yang et al., 2023b) performs remarkably better than VideoCLIP (Xu et al., 2021) in terms of DTW and OTAM, as it is trained to explore the global temporal context. However, all these methods suffer from noisy correspondence in the temporal alignment. In contrast, our proposed robust optimal transport framework explicitly overcomes multi-granularity noisy correspondence. Specifically, our method effectively improves the performance of all measurements by a large margin (+ 1% Cap. Avg., 5.2% DTW, and 4% OTAM in terms of R@1), indicating that our method learns better temporal information. ii) *Background Kept*: As shown in Table 2, compared with the *Background Removed* results, the recall of all methods dropped as the irrelevant information in the background can distract the video features. Nevertheless, our proposed method consistently outperformed VideoCLIP and TempCLR, even under such challenging conditions.

## 4.2 EVALUATION ON DIVERSE DOWNSTREAM TASKS

To verify the generalization of our method, we conduct experiments on three downstream tasks with four datasets described below.

**Text-to-Video retrieval (clip level).** This task aims to find a corresponding video clip given a query caption. We use YouCookII (Zhou et al., 2018) and MSR-VTT (Xu et al., 2016) to evaluate the transferability of our method. MSR-VTT (Xu et al., 2016) is a well-known retrieval benchmark containing 10,000 short videos with 20 captions each. Following Xu et al. (2021), we utilize the 1,000 clip-caption test pairs for evaluation. For YouCookII, we use 3,350 clip-caption pairs as introduced in Section 4.1.

As shown in Table 3, our method achieves remarkable improvement over state-of-the-art methods on YouCookII. On MSR-VTT (Table 5), our method shows solid improvements especially about 1.9% R@5 and 1.6% R@10 zero-shot improvement compared with VideoCLIP. After fine-tuning, our method still reaches state-of-the-art R@1. Here we include SupportSet (Patrick et al., 2021) and Frozen (Bain et al., 2021) for completeness, while they use different pre-training data such as 65 million Instagram videos (Ghadiyaram et al., 2019), 2.5 million WebVid videos (Bain et al., 2021) and 3 million Google Conceptual Captions (Sharma et al., 2018). The results in this clip-caption retrieval experiment indicate that our method not only improves the global temporal information (long video retrieval as shown in Section 4.1), but also facilitates clip-level representation learning.

Table 3: Clip-caption retrieval on YouCookII.

| Approach | Feature | R@1 | R@5 | R@10 |
|---|---|---|---|---|
| ActBERT (Zhu & Yang, 2020) | R101+Res3D | 9.6 | 26.7 | 38.0 |
| MIL-NCE (Miech et al., 2020) | S3D-G | 15.1 | 38.0 | 51.2 |
| MCN (Chen et al., 2021) | R152+RX101 | 18.1 | 35.5 | 45.2 |
| TACo (Yang et al., 2021a) | S3D-G | 19.9 | 43.2 | 55.7 |
| VT-TWINS (Ko et al., 2022) | S3D-G | 9.7 | 27.0 | 38.8 |
| MMFT (Shvetsova et al., 2022) | S3D-G | 19.8 | 42.9 | 55.1 |
| TAN (Han et al., 2022) | S3D-G | 20.1 | 45.5 | 59.5 |
| VideoCLIP (Xu et al., 2021) | S3D-G | 22.7 | 50.4 | 63.1 |
| TempCLR (Yang et al., 2023b) | S3D-G | 23.3 | 51.0 | **64.5** |
| Norton (Ours) | S3D-G | **24.2** | **51.9** | 64.1 |

Table 4: Action segmentation on COIN.

| Approach | Frame Accuracy |
|---|---|
| VAVA (Liu et al., 2022b) | 47.3 |
| ActBERT (Zhu & Yang, 2020) | 57.0 |
| Drop-DTW (Dvornik et al., 2021) | 59.6 |
| MIL-NCE (Miech et al., 2020) | 61.0 |
| ClipBERT (Lei et al., 2021) | 65.4 |
| TACo (Yang et al., 2021a) | 68.4 |
| VideoCLIP (Xu et al., 2021) | 68.7 |
| TempCLR (Yang et al., 2023b) | 68.7 |
| Norton (Ours) | **69.8** |

Table 5: Text-to-video retrieval on MSR-VTT.

| Superivsed | R@1 | R@5 | R@10 |
|---|---|---|---|
| SupportSet (Patrick et al., 2021) | 30.1 | 58.5 | 69.3 |
| Frozen (Bain et al., 2021) | 31.0 | 59.5 | 70.5 |
| MMFT (Shvetsova et al., 2022) | 23.7 | 52.1 | 63.7 |
| VideoCLIP (Xu et al., 2021) | 30.9 | 55.4 | **66.8** |
| TempCLR (Yang et al., 2023b) | 30.6 | 55.1 | 65.5 |
| Norton (Ours) | **31.2** | **55.7** | **66.8** |

| Zero-shot | R@1 | R@5 | R@10 |
|---|---|---|---|
| SupportSet (Patrick et al., 2021) | 8.7 | 23.0 | 31.1 |
| Frozen (Bain et al., 2021) | 23.2 | 44.6 | 56.6 |
| MIL-NCE (Miech et al., 2020) | 9.9 | 24.0 | 32.4 |
| MMFT (Shvetsova et al., 2022) | 9.9 | 24.0 | **32.6** |
| VT-TWINS (Ko et al., 2022) | 9.4 | 23.4 | 31.6 |
| VideoCLIP (Xu et al., 2021) | 10.4 | 22.2 | 30.0 |
| TempCLR (Yang et al., 2023b) | 10.1 | 22.2 | 29.4 |
| Norton (Ours) | **10.7** | **24.1** | 31.6 |

Table 6: VideoQA on MSR-VTT.

| Superivsed | Accuracy |
|---|---|
| EITanque (Kaufman et al., 2017) | 65.5 |
| MLB (Kim et al., 2016) | 76.1 |
| JSFusion (Yu et al., 2018) | 83.4 |
| ActBERT (Zhu & Yang, 2020) | 85.7 |
| ClipBERT (Lei et al., 2021) | 88.2 |
| MERLOT (Zellers et al., 2021) | 90.9 |
| VideoCLIP (Xu et al., 2021) | 92.1 |
| TempCLR (Yang et al., 2023b) | 92.2 |
| Norton (Ours) | **92.7** |

| Zero-shot | Accuracy |
|---|---|
| VideoCLIP (Xu et al., 2021) | 73.9 |
| TempCLR (Yang et al., 2023b) | 74.4 |
| Norton (Ours) | **77.1** |

**VideoQA.** We conduct the multiple choice VideoQA experiment on MSR-VTT (Yu et al., 2018). Given a video query and some candidate textual answers (5 on average), the task is to find the one that fits the query out of possible candidates. As shown in Table 6, our method outperforms the counterparts with +2.7% in terms of zero-shot accuracy and achieves 0.5% improvements after finetuning, showing the superiority of our method.

**Action Segmentation.** This task assumes that each video is associated with various actions. The goal is to determine the specific action for each second, which requires fully exploring the temporal dependencies. We use the long video dataset COIN (Tang et al., 2019) to evaluate the action segmentation performance of our method. COIN contains 11,827 videos (476 hours) in total where each video is labeled with 3.91 action segments on average, according to 778 candidate segment labels. Following Xu et al. (2021), we apply a one-layer classification head on top of the visual encoder to classify the action label. We report the frame-wise accuracy using the evaluation protocol of Xu et al. (2021); Miech et al. (2020). As shown in Table 4, our method outperforms all baselines.

Table 7: **Ablation experiments** evaluated on YouCookII, where "Clip" is short for clip-caption retrieval, "Video" for video-paragraph retrieval, "B" for video backgrounds, and "FNE" for faulty negative exploitation. We report the DTW measurement for video-paragraph retrieval.

| Basic Setting | | | | Clip | | Video (w/o B) | | Video (w B) | |
|---|---|---|---|---|---|---|---|---|---|
| Model | FNE | Soft-max $\alpha$ | APB $p$ | R@1 | R@5 | R@1 | R@5 | R@1 | R@5 |
| VideoCLIP (Xu et al., 2021) | – | – | – | 22.7 | 50.4 | 56.0 | 89.9 | 55.7 | 93.1 |
| TempCLR (Yang et al., 2023b) | – | – | – | 23.3 | 51.0 | 83.5 | 97.2 | 70.4 | 93.8 |
| A (w/o $\mathcal{L}_{video}$) | | – | – | 22.8 | 50.1 | 56.7 | 89.0 | 56.4 | 91.8 |
| B (w/o $\mathcal{L}_{video}$) | ✓ | – | – | 23.4 | 50.8 | 63.3 | 93.3 | 65.1 | 92.4 |
| C | ✓ | Mean average | – | 23.1 | 50.1 | 84.2 | 97.3 | 74.3 | **94.7** |
| D | ✓ | (Yao et al., 2022) | – | 23.5 | 50.5 | 86.9 | 98.6 | 74.1 | 94.6 |
| E | ✓ | 0.1 | – | 23.8 | 51.7 | 88.1 | 98.6 | 74.2 | **94.7** |
| F | ✓ | 0.2 | – | **24.0** | **51.8** | 88.2 | 98.6 | 74.9 | 94.4 |
| G | ✓ | 1 | – | **24.0** | **51.8** | 88.4 | 98.8 | 75.2 | **94.7** |
| H | ✓ | 1 | 10% | **24.2** | 51.8 | 88.4 | **98.8** | 75.9 | 94.9 |
| I | ✓ | 1 | 50% | **24.2** | 51.9 | 88.4 | 98.6 | 75.9 | 94.9 |
| J (Norton) | ✓ | 1 | 30% | **24.2** | **51.9** | **88.7** | **98.8** | **76.1** | **95.0** |

## 4.3 Ablation Study on the Proposed Methods

In this section, we investigate the effects of our design choices and discuss the results in Table 7.

**Effect of Faulty Negative Exploitation.** In model-{A,B}, we tackle the issue of faulty negatives in clip-caption contrastive learning through the correction of optimal transport. This strategy not only improves the performance of clip-caption retrieval but also enhances the temporal ability.

**Effect of OT in Temporal Learning.** In model-C, we utilize vanilla optimal transport to measure the distance between sequences where the clip/caption representation is obtained by averaging the frame/word embeddings. As shown, model-C achieves comparable performance to TempCLR and even outperforms TempCLR in retrieval tasks involving backgrounds.

**Effect of Fine-grained Alignment.** In model-{D,E,F,G}, we investigate the effect of fine-grained alignment by varying the weight of the log-sum-exp approximation. We also compare our approach with Yao et al. (2022) which selects the most important token for fine-grained alignment. The comparison demonstrates that our strategy outperforms Yao et al. (2022), supporting our claim that focusing on more crucial words/frames yields better fine-grained measurements in video understanding. When the weight $\alpha$ tends towards 0, the log-sum-exp approximation approximates the maximum, resulting in the selection of the most relevant words/frames. The comparison between model-{E,F,G} shows that a larger $\alpha$ leads to better performance, further validating our assumption that focusing on more important tokens would enhance performance.

**Effect of Alignable Prompt Bucket.** In model-{H,I,J}, we integrate the prompt bucket into the optimal transport framework and vary the value of $p$ to be the bottom 10%, 30%, and 50% similarity between the original aligned clips and captions. We observe that the use of APB results in a clear performance improvement for video-paragraph retrieval with background, and setting the value of $p$ to the bottom 30% similarity is an effective choice.

## 5 Conclusion

Learning temporal correlations in long-form videos is prohibitively expensive in terms of the hardware required. To address this, we propose Norton, a noise robust temporal optimal transport to estimate the sequence distance that can be easily extended and scaled to larger datasets with minimal computational cost. Notably, our unified optimal transport solution resolves the noisy correspondence problem at both frame-word and clip-caption levels. Extensive experiments demonstrate that our method not only captures long-term temporal dependencies but also facilitates clip-level representation learning. In the future, we plan to extend our method to address noisy correspondence for more modalities as videos typically include visual, textual, and audio content.

ACKNOWLEDGMENTS

This work was supported in part by NSFC under Grant U21B2040, 62176171; and in part by the Fundamental Research Funds for the Central Universities under Grant CJ202303.

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

APPENDIX

In this supplementary material, we present:

## A    DETAILS OF PRE-TRAINING

Following mainstream VLP works (Miech et al., 2020; Xu et al., 2021; Yang et al., 2023b; Han et al., 2022), we use the instructional videos HowTo100M (Miech et al., 2019) for pre-training. Below, we provide an overview of the network architecture, data sampling, and training setting.

**Architecture.** We adopt dual Transformer encoders (Devlin et al., 2018) for processing video clips and captions, separately. Specifically, the video encoder consists of a 6-layer Transformer, while the text encoder consists of a 12-layer Transformer. For each video clip, we use HowTo100M pre-trained S3D (Miech et al., 2020) to extract one video token per second at 30 fps. For each text, we obtain word tokens via embedding lookup as in BERT (Devlin et al., 2018). The video tokens and text tokens are then separately passed through the video and text Transformer encoders to obtain frame and word representations, respectively. As the quality of the representations plays a crucial role in temporal learning, we initialize our network with VideoCLIP checkpoint (Xu et al., 2021) due to limited computation resources, following the same setting of TempCLR (Yang et al., 2023b). Experimental results demonstrate that our method significantly improves VideoCLIP's performance on various long and short video tasks, with only 1 GPU day of post-training.

**Data sampling.** We follow the sampling strategy of VideoCLIP (Xu et al., 2021) as below:

1. Sample a text caption with $8 \sim 32$ tokens by merging the timestamps of the raw captions. This is done because sampling a video clip first may not have a corresponding caption nearby;

2. Sample a timestamp within the boundary of the caption as the center for a video clip;

3. Grow a video clip with random duration ($3 \sim 16$ seconds) from this center timestamp.

We sample 16 clips/captions from each HowTo100M video and form the long video sequence with consecutive 8 clips/captions. The batch size is set to 64 videos, resulting in 128 ($64 \times 16/8$) video sequences in total for video-paragraph contrastive learning in a mini-batch.

**Training setting.** We implement our method in PyTorch 1.11.0 (Paszke et al., 2019) and conduct all experiments on the Red Hat 6.4.0-1 OS. We train the network for 10 epochs with fp16 precision, which takes approximately 1 A100 GPU day. We use Adam optimizer (Kingma & Ba, 2014) with the learning rate of $1e^{-5}$ to optimize the network. Each training batch consisted of 64 videos, each paired with 16 corresponding clips and captions. We set the balanced weight $\lambda$ between clip and video loss to 0.1. The log-sum-exp parameter $\alpha$ and the faulty negative exploitation $\beta$ are set to 1 and 0.3, respectively. We run 50 steps of the Sinkhorn algorithm and set the entropy $\varepsilon$ to 0.1 and 1 for calculating the optimal transport in $\mathcal{L}_{\text{video}}$ and $\mathcal{L}_{\text{clip}}$, respectively.

To compute clip-caption loss $\mathcal{L}_{\text{clip}}$, we derive clip and caption representations through average pooling on the token embeddings of frames and words, respectively. For video-paragraph loss $\mathcal{L}_{\text{video}}$, we enhance the average pooling similarity by incorporating the proposed fine-grained similarity measure. For downstream tasks such as retrieval and QA, we maintain computational efficiency by averaging the embeddings of frames and words as the clip and caption representations, respectively.

## B    DERIVATION OF THE SINKHORN-KNOPP ITERATION

In this section, we briefly introduce the derivation of the Sinkhorn algorithm (Cuturi, 2013) for calculating the optimal transport distance. Given the similarity matrix $\mathbf{S} \in \mathbb{R}^{n \times m}$ where $[\mathbf{S}]_{a,b}$ measures the similarity between video clip $\mathbf{v}_a$ and caption $\mathbf{t}_b$, optimal transport aims to maximize the expectation of the global similarity through,

$$\max_{\mathbf{Q} \in \mathcal{Q}} \quad \langle \mathbf{Q}, \mathbf{S} \rangle = \mathrm{tr}(\mathbf{Q}^\top \mathbf{S}) = \sum_{a=1}^{n} \sum_{b=1}^{m} [\mathbf{Q}]_{a,b} \cdot [\mathbf{S}]_{a,b} \tag{9}$$
$$\text{s.t.} \quad \mathcal{Q} = \left\{ \mathbf{Q} \in \mathbb{R}_+^{n \times m} \mid \mathbf{Q} \mathbf{1}_m = \boldsymbol{\mu}, \mathbf{Q}^\top \mathbf{1}_n = \boldsymbol{\nu} \right\}.$$

where probability vectors $\boldsymbol{\mu}, \boldsymbol{\nu}$ denote the amount of mass that could transport from $\mathbf{v}$ to $\mathbf{t}$ (Wang et al., 2022a). If each clip in video $\mathbf{V}$ or caption in paragraph $\mathbf{T}$ is sampled independently from a distribution, the weights can be set equally, $i.e.$, $\boldsymbol{\mu} = \frac{1}{n} \mathbf{1}_n$ and $\boldsymbol{\nu} = \frac{1}{m} \mathbf{1}_m$.

Note that Eq. (9) is a standard linear programming problem and can be solved in polynomial time (around $O(n^3 \log n)$). However, considering the high volume of data points, common linear programming solvers can be time-consuming. To overcome this limitation, Cuturi (2013) investigates a fast approximation version of this optimization by adding an entropy regularization term,

$$\max_{\mathbf{Q} \in \mathcal{Q}} \quad \langle \mathbf{Q}, \mathbf{S} \rangle + \varepsilon H(\mathbf{Q}) \tag{10}$$

where $H(\mathbf{Q}) = -\sum_{ab} [\mathbf{Q}]_{a,b} \log [\mathbf{Q}]_{a,b}$ is derived from an optimization perspective and it makes the objective function smoothing and convex, allowing for efficient computation. Let $\mathcal{L}(\mathbf{Q}, \boldsymbol{u}, \boldsymbol{v})$ be the Lagrangian of Eq. (10) with dual multipliers $\boldsymbol{u} \in \mathbb{R}^n, \boldsymbol{v} \in \mathbb{R}^m$,

$$\mathcal{L}(\boldsymbol{Q}, \boldsymbol{u}, \boldsymbol{v}) = \langle \mathbf{Q}, \mathbf{S} \rangle + \varepsilon H(\mathbf{Q}) + \boldsymbol{u}^\top (\mathbf{Q} \mathbf{1}_L - \boldsymbol{\mu}) + \boldsymbol{v}^\top (\mathbf{Q}^\top \mathbf{1}_n - \boldsymbol{\nu}), \tag{11}$$

Since the original optimization problem is convex, the solution must satisfy the Karush-Kuhn-Tucker (KKT) conditions. Therefore, by taking the partial derivative of the Lagrangian in Eq.(11) with respect to $[\mathbf{Q}]_{a,b}$, we obtain the following equation:

$$\frac{\partial \mathcal{L}(\mathbf{Q}, \boldsymbol{u}, \boldsymbol{v})}{\partial [\mathbf{Q}]_{a,b}} = [\mathbf{S}]_{a,b} - \varepsilon \left( \log \left( [\mathbf{Q}]_{a,b} \right) + 1 \right) + \boldsymbol{u}_a + \boldsymbol{v}_b = 0 \tag{12}$$

For any couple $(a, b)$,

$$(\partial \mathcal{L} / \partial [\mathbf{Q}]_{a,b} = 0) \Rightarrow [\mathbf{Q}]_{a,b} = e^{-\frac{1}{2} + \frac{\boldsymbol{u}_a}{\varepsilon}} \ e^{\frac{[\mathbf{S}]_{a,b}}{\varepsilon}} \ e^{-\frac{1}{2} + \frac{\boldsymbol{v}_b}{\varepsilon}}. \tag{13}$$

Therefore, solving Eq. (10) equals to finding the dual multipliers $\boldsymbol{u}$ and $\boldsymbol{v}$, which is also equivalent to get another two scaling coefficients vectors $\boldsymbol{\kappa}_1 \in \mathbb{R}^n, \boldsymbol{\kappa}_2, \in \mathbb{R}^m$ such that,

$$[\boldsymbol{\kappa}_1]_a = e^{-\frac{1}{2} + \frac{\boldsymbol{u}_a}{\varepsilon}} \quad \text{and} \quad [\boldsymbol{\kappa}_2]_b = e^{-\frac{1}{2} + \frac{\boldsymbol{v}_b}{\varepsilon}}. \tag{14}$$

These scaling coefficients are used to compute the optimal transport matrix $\mathbf{Q}$ as a normalized exponential matrix form,

$$\mathbf{Q}^* = \mathrm{Diag}(\boldsymbol{\kappa}_1) \exp \left( \mathbf{S} / \varepsilon \right) \mathrm{Diag}(\boldsymbol{\kappa}_2). \tag{15}$$

Recall $\mathbf{Q}^*$ meets the constraints in Eq. (9) that,

$$\mathbf{Q}^* \mathbf{1}_m = \mathrm{Diag}(\boldsymbol{\kappa}_1) \exp \left( \mathbf{S} / \varepsilon \right) \boldsymbol{\kappa}_2 = \boldsymbol{\mu}, \quad \mathbf{Q}^{*\top} \mathbf{1}_n = \mathrm{Diag}(\boldsymbol{\kappa}_2) \exp \left( \mathbf{S}^\top / \varepsilon \right) \boldsymbol{\kappa}_1 = \boldsymbol{\nu}, \tag{16}$$

which gives rise to an alternative coordinate descent algorithm, known as the Sinkhorn-Knopp fixed point iteration (Cuturi, 2013), that updates the scaling coefficients as follows:

$$\boldsymbol{\kappa}_1 \leftarrow \boldsymbol{\mu} ./ \left( \exp \left( \mathbf{S} / \varepsilon \right) \boldsymbol{\kappa}_2 \right), \quad \boldsymbol{\kappa}_2 \leftarrow \boldsymbol{\nu} ./ \left( \exp \left( \mathbf{S}^\top / \varepsilon \right) \boldsymbol{\kappa}_1 \right). \tag{17}$$

Empirically, running 50 steps is often sufficient to obtain a satisfactory alignment result. Finally, we get optimal transport distance through $\langle \mathbf{Q}^*, \mathbf{S} \rangle$.

Table 8: **Training time per epoch.** 'f' denotes the sampled frame for a video clip. We use the time cost of clip-caption contrastive learning (Line 1) as the base value for comparison in the third column. The default setting is marked in gray.

| Line | Approach | Time Cost |
|---|---|---|
| 1 | Clip-caption Contrast (16f) | 87 min ($\times 1.000$) |
| 2 | + Faulty Negative Exploitation | 92 min ($\times 1.057$) |
| 3 | + Video-paragraph Contrast (16f$\times$8) | 142 min ($\times 1.632$) |
| 4 | + Fine-grained Soft-maximum Operator (16f$\times$8) | 146 min ($\times 1.678$) |
| 5 | Clip-caption Contrast (32f) | 172 min ($\times 1.977$) |
| 6 | Sinkhorn iteration in $\mathcal{L}_{\text{clip}}$ | 2.4 min ($\times 0.027$) |
| 7 | Sinkhorn iteration in $\mathcal{L}_{\text{video}}$ | 2.6 min ($\times 0.029$) |

## C    TRAINING EFFICIENCY DISCUSSION

Most existing temporal learning methods (Han et al., 2022; Zeng et al., 2023b) directly model long videos using the video Transformer encoder. However, the complexity of the Transformer (Devlin et al., 2018; Vaswani et al., 2017) is approximately $O(t^2)$, where $t$ is the number of the video frames. Consequently, these methods require significant computational resources to model lengthy videos. In contrast, our approach utilizes optimal transport to estimate the sequence distance between short video clips and captions in a late fusion manner, thereby alleviating the need to encode entire long videos. Although the complexity of the Sinkhorn algorithm is directly proportional to the number of video clips, captions, and Sinkhorn iterations, this late computation is negligible compared to the computation of the deep network.

Table 8 presents the training time for different settings on a single A100 GPU. "16f" indicates that we sample video clips up to 16 seconds. "16f$\times$8" denotes that we employ OT to measure the distance between 8 clips and 8 captions, resulting in a sequence length increase to 128. For contrastive learning in Lines 1 and 5, we average the token embeddings of frames/words as the clip/caption representation following Xu et al. (2021). As shown, the proposed faulty negative exploitation (Line 2) and fine-grained operator (Line 4) only require a small amount of time compared with Lines 1 and 3. This is because our fine-grained operator and optimal transport both operate in a late interaction mechanism, which is only conducted on the final output of the encoder.

When extending the video length to 32 frames (Line 5), the training time increases from 87 minutes (Line 1) to 172 minutes (approximately $\times 1.98$). This experiment aims to simulate temporal learning methods that encode the entire long video into a single sequence (Zeng et al., 2023b; Han et al., 2022). In contrast, our method (Line 4) requires a smaller amount of time (146 minutes) while still being capable of embedding videos with 128 frames. We further evaluate the time cost of Sinkhorn iteration per epoch in Lines 6 and 7. Compared to the forward and backward passes of the network, the computation of the Sinkhorn iterations is minimal.

## D    ROBUSTNESS ON NOISY CORRESPONDENCE

In this section, we evaluate the effectiveness of different methods against noisy correspondence through visual-textual alignment experiments on the HTM-Align dataset (Han et al., 2022). HTM-Align is a subset of the HowTo100M dataset, consisting of 80 videos with 49K sentences that have been **manually annotated to rectify the alignment** in the presence of noisy correspondence. The annotators have two main tasks: i) determining if a sentence from ASR is visually related to the video, and ii) adjusting the start & end timestamps to accurately cover the visual content if the sentence is related.

After training the models on the HowTo100M dataset, we evaluated their performance on this alignment task to assess their ability to handle noise. We report the Recall metrics for this alignment task. Specifically, for a **misaligned sentence**, if its most closely matched video frame falls into the ground-truth segment annotated by the human, it is counted as a successful recall. The Recall scores are averaged across all the text segments.

For a fair comparison, the maximum number of video frames is set to 32 for Han et al. (2022); Xu et al. (2021); Yang et al. (2023b) and our method. We also include the 64 frame version of TAN (Han et al., 2022) for completeness. We use a sliding window approach to calculate the similarity between video frames and sentences with a window size of 32 seconds and a step size of 8 seconds. We averaged the similarity scores for overlapping visual tokens from multiple windows. As shown in Table 9, CLIP exhibits inferior performance, possibly because it has only been trained on images and lacks the ability to capture video dynamics. In contrast, our method outperforms VideoCLIP and TempCLR, providing evidence that our approach is not prone to fit noisy correspondence.

Table 9: Alignment results on the HTM-Align datasets.

| Approach | Recall |
|---|---|
| CLIP (ViT-B/32) (Radford et al., 2021) | 17.5 |
| MIL-NCE (Miech et al., 2020) | 34.2 |
| TAN (Han et al., 2022) - 32 frame | 41.1 |
| TAN (Han et al., 2022) - 64 frame | 49.2 |
| VideoCLIP (Xu et al., 2021) | 44.4 |
| TempCLR (Yang et al., 2023b) | 44.1 |
| Norton (Ours) | **46.9** |

## E  APPLICATIONS AND POTENTIAL IMPLICATIONS

**Application scenarios.** Norton is a representation learning method that exhibits versatility across various tasks including video retrieval, video QA, and classification, as confirmed by our experiments. A notable strength of Norton lies in its ability to effectively address the common challenge of noisy correspondence, particularly in **uncurated instructional videos**. This adaptability allows Norton to be implemented in diverse scenarios without necessitating meticulous video curation. For instance, Norton proves effective in tasks such as long video retrieval or classification for various content genres like movies, education videos, and cooking tutorials. It's also essential to acknowledge that Norton is tailored for representation learning and may exhibit suboptimal performance in tasks focused on content generation, such as video captioning.

**Potential implications.** This paper delves into two challenging problems in video understanding, namely, long video learning and noisy correspondence learning. In addressing the former, where computational constraints have limited prior works, our proposed efficient solution may spark increased interest in long video understanding tasks. Regarding the latter, the noisy correspondence problem (**mismatched data pairs**) has garnered attention in diverse multi-modal applications, extending **beyond video-text domains** to encompass challenges in image-text retrieval (Huang et al., 2021; Qin et al., 2022; 2023; Han et al., 2023; Yang et al., 2023a), cross-modal generation (Li et al., 2022), person re-identification (Yang et al., 2022), and graph matching (Lin et al., 2023). Our work has the potential to attract increased attention to the broader spectrum of noisy correspondence challenges across various domains.

## F  CHALLENGES IN FUTURE WORKS

**Multi-modal scenarios.** Our approach introduces an optimal transport solution to address the noisy correspondence between bi-modalities in videos and text. However, as videos inherently encompass visual, textual, and audio content (Shvetsova et al., 2022; Yang et al., 2024), the noisy correspondence challenge might extend across multiple modalities. Addressing multi-modal noisy correspondence using optimal transport presents an open challenge, given the quadratic growth in combinations concerning the number of modalities. We acknowledge this limitation and plan to extend our method to effectively tackle multi-modal noisy correspondence, exploring these scenarios in future work.

**Utilization of Noise.** In this paper, we employ the prompt bucket to directly filter out irrelevant clips and captions during sequential alignment, attempting to mitigate the influence of noisy correspondence. However, an intriguing question arises regarding whether these noisy samples could be

utilized as an incentive for training (Li, 2022). Exploring the possibility of generating associated text for unalignable video clips using large multimodal models (LMMs), *e.g.*, LLaVA (Liu et al., 2023), BLIP-2 (Li et al., 2023) and GPT-4V(ision) (OpenAI, 2023), could open up a novel avenue for exploration and improvement in future research endeavors.

## G   VISUALIZATION OF RE-ALIGNMENT FOR YOUTUBE VIDEOS

In this section, we present the visualization of the optimal transport assignment $\mathbf{Q}$ to demonstrate the robustness of our method. Specifically, we compared our proposed Norton with the Dynamic Time Warping and vanilla optimal transport. As shown in Fig. 3c, the vanilla OT falsely aligns the meaningless text "It's a tense moment" to some irrelevant video clips, because OT requires exact mapping between each source instance and the targets. In contrast, as depicted in Fig. 3d, our method successfully filters out the semantically irrelevant captions with the help of the proposed Alignable Prompt Bucket. Moreover, as shown in Fig. 3b, DTW erroneously aligns video clips to multiple captions and fails to address the issue of irrelevant captions. In a word, the visualization illustrates that our Norton outperforms DTW and vanilla OT in aligning the clips with captions in the presence of noisy correspondence.

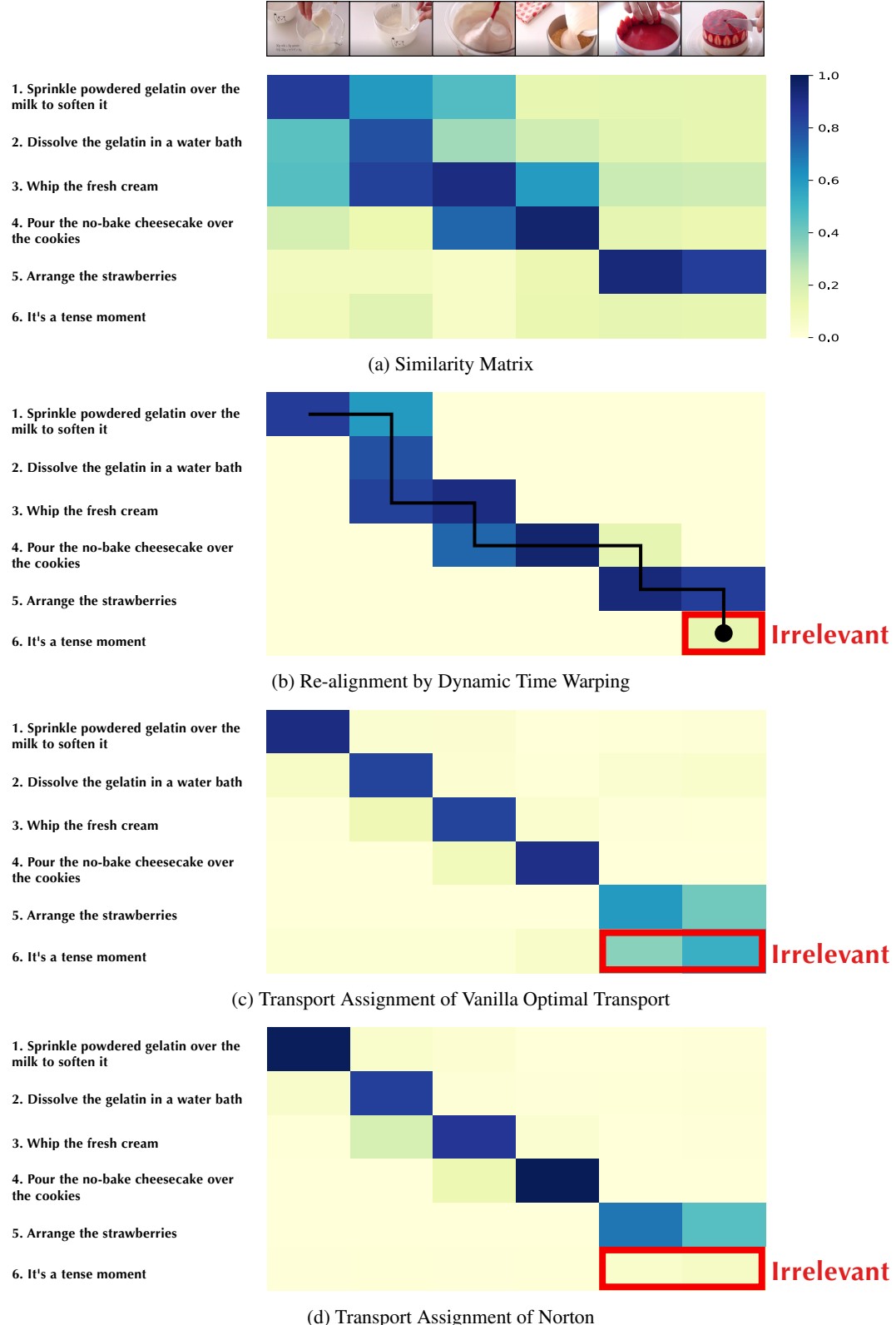

Figure 3: Visualization of the re-alignment.

