# OpenReview forum: "Multi-granularity Correspondence Learning from Long-term Noisy Videos"
_ICLR.cc/2024/Conference — ICLR 2024 oral_

### Official Review · Reviewer_83JB · 2023-10-29

**Soundness:** 4 excellent
**Presentation:** 4 excellent
**Contribution:** 4 excellent
**Rating:** 8
**Confidence:** 5

**Summary:**

This paper proposes a novel method for learning long-term temporal correspondence from noisy instructional videos. The main idea is to use optimal transport (OT) to measure the sequence similarity between video clips and captions, and address the multi-granularity noisy correspondence (MNC) problem at both coarse and fine levels. The paper also introduces several techniques to enhance the OT framework, such as a soft-maximum operator, an alignable prompt bucket, and a faulty negative exploitation strategy. The paper evaluates the proposed method on various downstream tasks, such as video-paragraph retrieval, videoQA, and action segmentation, and shows that it outperforms existing state-of-the-art methods.

**Strengths:**

(1) The paper tackles an important and challenging MNC problem of learning long-term temporal dependency from noisy video-text data, which has many potential applications in video understanding. Although temporal misalignment has been explored in TAN [1], they only consider the noisy correspondence from a course grained sentence level. While this paper proposes a novel and efficient method based on optimal transport, which can handle the multi-granularity noisy correspondence problem in a unified framework and outperform previous work TAN. The paper also provides empirical evidence to support the proposed method.
(2) The paper introduces several innovative components to enhance the optimal transport framework, such as the soft-maximum operator for fine-grained alignment, the alignable prompt bucket for filtering out irrelevant clips or captions, and the faulty negative exploitation for improving clip representation.
(3) The paper conducts extensive experiments on diverse downstream tasks and datasets and demonstrates that the proposed method achieves remarkable improvements over existing state-of-the-art methods. The paper also performs ablation studies to analyze the impact of different design choices.

**Weaknesses:**

(1) The explanation of why using optimal transport effectively learns video-paragraph similarity in the paper could be more explicit. Specifically, when the similarity calculated by the S matrix might not be accurate in the early stages of model training, it is important to understand how the training objective, which aims to maximize the similarity of ⟨Q,S⟩, prevents misleading the model. Additional clarification on this point would be beneficial.
(2) The paper does not compare the proposed method with other methods that use optimal transport for sequence alignment, such as Su & Hua (2017) [2]. It would be interesting to see how the proposed method differs from these methods in terms of performance and efficiency.

**Questions:**

(1) How do you choose the value of p of clip-caption pairs for the alignable prompt bucket? Is it fixed per epoch or adaptive to different batches? How sensitive is the performance to different values of p?
(2) How do you deal with clips or captions that have different lengths as you choose a random window size for clip and caption sampling? Do you perform any preprocessing or padding on the clips or captions? How does this affect the optimal transport computation?
[1] Tengda Han, Weidi Xie, and Andrew Zisserman. Temporal alignment network for long-term video. In Proceedings of the IEEE/CVF Conference on Computer Vision and Pattern Recognition, 2022.
[2 ] Bing Su and Gang Hua. Order-preserving wasserstein distance for sequence matching. In Proceedings of the IEEE conference on computer vision and pattern recognition, pp. 1049–1057, 2017.

---

> ### Author Response · Authors · 2023-11-17
> **The Response to Reviewer 83JB (Part 1/2)**
>
> ##
>
> Thanks a lot for reviewing our paper and giving us valuable suggestions.  We will answer the questions one by one.
>
> > ***Question 1:** When the similarity calculated by the $\mathbf{S}$ matrix might not be accurate in the early stages of model training, it is important to understand how the training objective, which aims to maximize the similarity of $\langle \mathbf{Q},\mathbf{S}\rangle$, prevents misleading the model.*
>
> To address potential inaccuracies in the similarity matrix $\mathbf{S}$ during video-paragraph loss computation, we have implemented a **warmup** strategy during the initial training phases to proactively mitigate this potential issue. Specifically, we initiate training by exclusively utilizing the clip-caption contrastive loss $\mathcal{L}_{\text{clip}}$. This initial focus on clip/caption pairs enhances the quality of their representations before introducing the video-paragraph contrastive loss.
>
> In our implementation, we build our network directly on top of the VideoCLIP checkpoint (trained with the clip-caption contrastive loss) and effectively reduce computational resources. Experimental results underscore the substantial performance improvement of our method over VideoCLIP in various long and short video tasks, achieved with only 1 GPU day of post-training.
>
> > ***Question 2:** The paper does not compare the proposed method with other methods that use optimal transport for sequence alignment, such as Su & Hua (2017). It would be interesting to see how the proposed method differs from these methods in terms of performance and efficiency.*
>
> Thanks for your suggestions. Su & Hua (2017) introduce an optimal transport method for sequence matching by incorporating two novel regularization terms. These terms, namely inverse difference moment regularization and KL divergence with a prior distribution regularization, aim to encourage transports to nearby instances and penalize alignments between distant instances.
>
> We have re-implemented Su & Hua (2017) in the context of video learning and evaluated it on the YouCookII dataset. In the table below, we present the results for clip-caption retrieval (marked as "Clip"), video-paragraph retrieval with video backgrounds (marked as "Video (w B)"), and video-paragraph retrieval without video backgrounds (marked as "Video (w/o B)"). Notably, our proposed method Norton outperforms Su & Hua (2017) by a significant margin in both short and long video retrieval tasks, with only a slight increase in time cost.
>
> | Method          |   Clip   |   Clip   | Video (w/o B) | Video (w/o B) | Video  (w B) | Video  (w B) | Time cost (per epoch) |
> | :-------------- | :------: | :------: | :-----------: | :-----------: | :----------: | :----------: | :-------------------: |
> |                 |   R@1    |   R@5    |      R@1      |      R@5      |     R@1      |     R@5      |        minute         |
> | Su & Hua (2017) |   23.3   |   50.3   |     84.4      |     97.2      |     74.3     |     94.3     |        **139**        |
> | Norton (Ours)   | **24.2** | **51.9** |   **88.7**    |   **98.8**    |   **76.1**   |   **95.0**   |          146          |
>
> It's essential to highlight that existing optimal transport works, including Su & Hua (2017), do not specifically focus on the alignment of video and text, which is the primary focus of our research. Beyond addressing traditional sequence alignment, we identify and address the fine-grained misalignment problem specific to video-text learning. Our approach introduces a novel soft-maximum operator that unifies multi-grained correspondence learning within the optimal transport framework, distinguishing our method from prior works.

---

> ### Author Response · Authors · 2023-11-17
> **The Response to Reviewer 83JB (Part 2/2)**
>
> > ***Question 3:** How do you choose the value of $p$ of clip-caption pairs for the alignable prompt bucket? Is it fixed per epoch or adaptive to different batches? How sensitive is the performance to different values of $p$?*
>
> The alignable prompt bucket (APB) is designed to filter out irrelevant clips and captions during coarse-grained clip-caption alignment. The value $p$ of the bucket functions as the **similarity margin** that distinguishes between alignable and unalignable clips and captions. In our implementation, we dynamically determine the value of $p$ as the bottom 30% similarity of the original aligned clip-caption pairs for **each batch**. This real-time adaptive approach ensures the relevance of the margin $p$ with the clip/caption pairs during training, responding to potential changes in alignment patterns per iteration.
>
> In ablation studies, we integrated the prompt bucket into the optimal transport framework and varied the value of $p$ as the bottom 10%, 30%, and 50% similarity between the original aligned clips and captions. The results presented in the table below demonstrate our method's robustness to different choices of $p$.
>
> | Method           |   Clip   |   Clip   | Video (w/o B) | Video (w/o B) | Video  (w B) | Video  (w B) |
> | :--------------- | :------: | :------: | :-----------: | :-----------: | :----------: | :----------: |
> |                  |   R@1    |   R@5    |      R@1      |      R@5      |     R@1      |     R@5      |
> | $p$=10%          | **24.2** |   51.8   |     88.4      |   **98.8**    |     75.9     |     94.9     |
> | $p$=50%          | **24.2** | **51.9** |     88.4      |     98.6      |     75.9     |     94.9     |
> | $p$=30% （ours） | **24.2** | **51.9** |   **88.7**    |   **98.8**    |   **76.1**   |   **95.0**   |
>
> > ***Question 4:** How do you deal with clips or captions that have different lengths as you choose a random window size for clip and caption sampling? Do you perform any preprocessing or padding on the clips or captions? How does this affect the optimal transport computation?*
>
> Our method could **adaptably** handle clips and captions with varying lengths without necessitating any preprocessing steps. Leveraging the proposed soft-maximum operator, we employ the log-sum-exp approximation to identify the crucial words or key frames for each frame or word, respectively. Following this, we average these soft-maximum similarities across all frames or words to establish the clip-caption similarity. This innovative approach ensures that the similarity between the clip and caption with different lengths can be effortlessly computed, facilitating seamless integration into the subsequent optimal transport sequential alignment process. In summary, the soft-maximum operator effectively addresses the challenge posed by variable lengths in clips and captions.

---

### Official Review · Reviewer_pP85 · 2023-10-30

**Soundness:** 4 excellent
**Presentation:** 4 excellent
**Contribution:** 4 excellent
**Rating:** 8
**Confidence:** 5

**Summary:**

This paper delves into the long-term video-text learning task and reveals a new problem named multi-granularity noisy correspondence (MNC), referring to both course-grained clip-caption misalignment and fine-grained frame-word misalignment. Clearly, such a problem would hinder temporal learning and video understanding. To address the MNC problem, the authors propose NOise Robust Temporal Optimal traNsport (Norton), which formulates the solutions to both course- and fine-grained NC into a unified optimal transport (OT) framework. On the one hand, Norton filters the irrelevant clips and captions using an alignable prompt bucket and realigns the asynchronous clip-caption pairs based on transport distance, contributing to robustness against course-grained NC. On the other hand, a soft-maximum operator is used to identify crucial words and keyframes so that the negative impact of fine-grained NC can be alleviated. The effectiveness of Norton is validated through extensive experiments on commonly used video-text tasks, including video retrieval, video QA, and action segmentation.

**Strengths:**

**Revealing a new problem**. This paper studies a new and practical challenge in the context of long-term video-text representation learning, namely, multi-granularity noisy correspondence (MNC). MNC encompasses both coarse-grained clip-caption misalignment and fine-grained frame-word misalignment, both of which hinder temporal learning and video comprehension. While some studies have been concentrated on addressing the coarse-grained clip-caption misalignment, as far as I know, there are no formal studies on the fine-grained NC for video-text learning. From this perspective, I think this paper would bring some new insights to the community.

**Novel approach**. To handle MNC and achieve robust long-term video-text learning, this paper proposes Norton, which formulates the solutions to both course- and fine-grained NC into a unified optimal transport (OT) framework. Norton first incorporates a token-wise soft-maximum operator to identify crucial words and keyframes within each clip-caption pair, so that the fine-grained NC could be eliminated. After that, Norton filters the irrelevant clips and captions using an alignable prompt bucket, and realigns the asynchronous clip-caption pairs based on transport distance, leading to robustness against course-grained NC.

**Good shape**. This paper is well-written and structured. Besides, the experiment designs are interesting and sufficient. Extensive experimental results validated the effectiveness of the proposed methods and the necessity of solving MNC problems.

**Weaknesses:**

- While the authors effectively illustrate the motivation in Fig. 1, the advantages of the proposed OT method over DTW require more elaboration. It is advisable to include further discussions to expound upon and clarify the claims made regarding the superiority of OT over DTW.
- The use of the stop-gradient operation in transport assignment Q is highlighted by the authors as a means to enhance the efficiency of their video-paragraph contrastive loss. However, the rationale behind this operation's efficacy is not entirely clear. To remedy this, additional discussions should be included to provide a more comprehensive explanation of why this operation is meaningful and how it improves efficiency.
- To enhance clarity, it is suggested that the authors highlight the second-best results alongside the primary results in each table. This practice can provide a useful point of reference for readers and facilitate a more comprehensive understanding of the findings.
- The results of DTW should be included as a baseline for comparison. This can help demonstrate the advancements made in the proposed methodology and provide a clearer context for the contributions of this work.

**Questions:**

The major concern is the lack of some claims, as highlighted in the weaknesses.

---

> ### Author Response · Authors · 2023-11-17
> **The Response to Reviewer pP85 (Part 1/2)**
>
> Thank you for acknowledging our efforts in revealing a new problem and presenting an innovative approach! We are prepared to address your questions one by one.
>
> > ***Question 1:** While the authors effectively illustrate the motivation in Fig. 1, the advantages of the proposed OT method over DTW require more elaboration. It is advisable to include further discussions to expound upon and clarify the claims made regarding the superiority of OT over DTW.*
>
> The advantages of our proposed Optimal Transport (OT) method over Dynamic Time Warping (DTW) are multi-faceted, as detailed below.
>
> 1. **Integration of multi-grained correspondence learning.** Our method tackles both coarse-grained and fine-grained misalignment within a unified OT framework from a fine-to-coarse perspective. In contrast, DTW predominantly focuses on the single-grained level.
> 2. **Suitability for non-monotonic alignment.** DTW aligns sequences following chronological order, limiting its ability to handle non-monotonic alignment cases. For instance, while DTW may align $\mathbf{t}_4$ with $\mathbf{v}_6$, it cannot align $\mathbf{t}_5$ with $\mathbf{v}_4$ and $\mathbf{v}_5$ simultaneously, as shown in Fig. 1. In contrast, our OT-based method is inherently flexible in addressing such scenarios.
> 3. **Handling of irrelevant misalignments.** DTW may erroneously align irrelevant clips and captions, such as aligning $\mathbf{t}_2$ with $\mathbf{v}_3$ in Fig. 1. Our method mitigates this challenge by introducing an alignment prompt bucket within the OT framework, effectively filtering out meaningless clips and captions.
> 4. **Robustness to sequence lengths.** OT demonstrates inherent robustness to variations in sequence lengths, which is a key distinction from DTW. The constraint terms on transport assignment $\mathbf{Q}$ in Eq. (2) ensure a fixed total mass (i.e., distance) regardless of the number of clips or captions. This characteristic makes our method well-suited for real-world applications with variational video lengths.
>
> These points collectively underline the superiority of our method over DTW in video-text learning and emphasize its robustness in handling diverse alignment challenges. Additionally, we placed **visual comparison results** between our method and DTW in Appendix G. We encourage reviewers to refer to the updated manuscript for further details.
>
>
>
>
>
> > ***Question 2:** The use of the stop-gradient operation in transport assignment Q is highlighted by the authors as a means to enhance the efficiency of their video-paragraph contrastive loss. However, the rationale behind this operation's efficacy is not entirely clear. To remedy this, additional discussions should be included to provide a more comprehensive explanation of why this operation is meaningful and how it improves efficiency.*
>
> In the derivation of the Sinkhorn-Knopp iteration (Appendix B), we observe the necessity for iterative matrix normalization to obtain the transport assignment $\mathbf{Q}$, defined as:
>
> $$ \mathbf{Q} =  \operatorname{Diag}(\boldsymbol{\kappa}_1) \exp \left({ \mathbf{S}}/{\varepsilon}\right) \operatorname{Diag}(\boldsymbol{\kappa}_2),$$
>
> with iteratively updated scaling vectors $\boldsymbol{\kappa}_1 \in \mathbb{R}^n$ and $\boldsymbol{\kappa}_2 \in \mathbb{R}^m$, calculated as
>
>  $$\boldsymbol{\kappa}_1 \leftarrow \boldsymbol{\mu} . /\left(\exp (\mathbf{S} / \varepsilon) \boldsymbol{\kappa}_2\right), ~\boldsymbol{\kappa}_2 \leftarrow \boldsymbol{\nu} . /\left(\exp \left(\mathbf{S}^{\top} / \varepsilon\right) \boldsymbol{\kappa}_1\right),$$
>
> where $\boldsymbol{\mu}=\frac{1}{n} \mathbf{1}_n \text { and } \boldsymbol{\nu}=\frac{1}{m} \mathbf{1}_m$ are the uniform probability distributions. Empirically, approximately 50 steps of the iterative updates are necessary to achieve a satisfactory assignment result. Retaining the gradient of $\mathbf{S}$ during the Sinkhorn iteration would introduce complexity to the backward pass of the video-paragraph contrastive loss. To maintain computational efficiency, we strategically choose to stop the gradient of the transport assignment $\mathbf{Q}$. This decision streamlines the optimization process and ensures its stability.
>
> > ***Question 3:** To enhance clarity, it is suggested that the authors highlight the second-best results alongside the primary results in each table.*
>
> Thanks. We have highlighted the second-best results with underlines and the best results with bold formatting in the manuscript following your advice.

---

> ### Author Response · Authors · 2023-11-17
> **The Response to Reviewer pP85 (Part 2/2)**
>
> ##
>
> > ***Question 4:** The results of DTW should be included as a baseline for comparison. This can help demonstrate the advancements made in the proposed methodology and provide a clearer context for the contributions of this work.*
>
> Thanks for your suggestion. It's worth noting that TempCLR (Yang et al., 2023) [ref.A] serves as a strong DTW baseline in video-text learning. TempCLR utilizes Dynamic Time Warping to measure sequential distances between video clips and captions. The methodology involves shuffling units in the positive sequence to sample negatives and enabling the contrasting of video with paragraphs to incorporate temporal correlation. Importantly, TempCLR also builds upon the VideoCLIP  (Xu et al., 2021) [ref.B], ensuring a fair comparison with our method.
>
> The table below presents retrieval results on the YouCookII dataset for clip-caption retrieval (marked as "Clip"), video-paragraph retrieval with video backgrounds (marked as "Video (w B)"), and video-paragraph retrieval without video backgrounds (marked as "Video (w/o B)"). Notably, our method significantly outperforms both VideoCLIP and TempCLR.
>
> | Method                      |   Clip   |   Clip   | Video (w/o B) | Video (w/o B) | Video  (w B) | Video  (w B) |
> | :-------------------------- | :------: | :------: | :-----------: | :-----------: | :----------: | :----------: |
> |                             |   R@1    |   R@5    |      R@1      |      R@5      |     R@1      |     R@5      |
> | VideoCLIP (Xu et al., 2021) |   22.7   |   50.4   |     56.0      |     89.9      |     55.7     |     93.1     |
> | TempCLR (Yang et al., 2023) |   23.3   |   51.0   |     83.5      |     97.2      |     70.4     |     93.8     |
> | Norton (Ours)               | **24.2** | **51.9** |   **88.7**    |   **98.8**    |   **76.1**   |   **95.0**   |
>
> [A] Yuncong Yang, Jiawei Ma, Shiyuan Huang, Long Chen, Xudong Lin, Guangxing Han, and Shih-Fu Chang. Tempclr: Temporal alignment representation with contrastive learning. In Proceedings of the International Conference on Learning Representations (ICLR), 2023.
>
> [B] Hu Xu, Gargi Ghosh, Po-Yao Huang, Dmytro Okhonko, Armen Aghajanyan, Florian Metze, Luke Zettlemoyer, and Christoph Feichtenhofer. Videoclip: Contrastive pre-training for zero-shot video-text understanding. In Proceedings of the 2021 Conference on Empirical Methods in Natural Language Processing, pp. 6787–6800, 2021.

---

### Official Review · Reviewer_TtfV · 2023-10-31

**Soundness:** 4 excellent
**Presentation:** 4 excellent
**Contribution:** 4 excellent
**Rating:** 8
**Confidence:** 4

**Summary:**

The paper presents an innovative approach Norton for video-language pre-training that learns long-term temporal dependencies between short video clips and captions. The paper addresses two challenges: 1) the high computational cost of modeling long videos and 2) the noisy correspondence between clips and captions due to asynchronous and irrelevant pairs. The paper uses a modified optimal transport framework to measure the sequence similarity between clips and captions, and to filter out the noisy pairs. The paper also exploits the faulty negative samples in contrastive learning to improve clip representation. The paper evaluates the method on video-paragraph retrieval, text-to-video retrieval, videoQA, and action segmentation tasks, and shows that it outperforms existing methods on various metrics. The paper also conducts ablation studies to analyze the impact of different components.

**Strengths:**

1.This paper is well-written, well-organized, and easy to follow.
2.The problem tackled by the authors is an interesting one - noisy correspondence, almost all video harvested from the web would include such cases and hinders the performance of large-scale multi-modal video learning. This challenge is inherent in the data and has received little attention in the literature. The paper explores this novel and important problem in depth.
3.The paper presents a unified OT framework that can efficiently and effectively solve the MNC problem at different levels of granularity. I commend the authors for providing a time cost table in the appendix which shows the efficiency of their method with various settings and verifies their claims.
This method's unique strength lies in using existing pre-training model and works in a self-bootstrapping capability. Note that directly pretraining a large multi-modal model is unpractical for the researchers not in company. This paper significantly improves the temporal ability of VideoCLIP without the need for additional models like DecemBert or TAN. This self-sufficiency significantly contributes to its enhanced scalability.

**Weaknesses:**

1.This paper does not provide specific numerical results on how well the proposed method handles noisy correspondence. It would be beneficial if the authors could provide more detailed numerical results or analysis on this aspect in their rebuttal or future work. For example, they could conduct experiments on synthetic noisy datasets to show the performance of their method. They could also compare their method with other methods that are designed to handle noisy correspondence and show how their method performs in comparison. This would provide more concrete evidence on the effectiveness of their method in handling noisy correspondence.
2.In certain scenarios, such as video-paragraph retrieval on YouCookII (as shown in Table 1) and in the context of ablation experiments (as indicated in Table 7), we observe encouraging indications of potential improvements. While in other experiments (as depicted in Table 2), there are mixed results across various metrics. Can you shed light on the reasons behind this disparity?
3.How does the proposed method compare with other OT-based methods like action sequence matching? What are the benefits or drawbacks of using OT for video-text learning? Please provide some comparisons and discussions.

**Questions:**

The primary queries for the rebuttal are predominantly derived from the "weaknesses" section outlined earlier. For instance, it would be highly appreciated if the authors could augment their experiments regarding noisy correspondence and offer more clarification on how their approach differs from other OT-based methods. Resolving these raised concerns will make the submission stronger and I vote for accepting this paper.

---

> ### Author Response · Authors · 2023-11-17
> **The Response to Reviewer TtfV (Part 1/2)**
>
> Thanks for your acknowledgment of our work! We will answer the questions one by one.
>
> > ***Question 1:** This paper does not provide specific numerical results on how well the proposed method handles noisy correspondence. It would be beneficial if the authors could provide more detailed numerical results or analysis on this aspect in their rebuttal or future work.*
>
> Thank you for the suggestions. To address the concerns regarding the noisy correspondence, we have conducted an evaluation using the HTM-Align dataset [ref.A] and included the results in Appendix D. HTM-Align is a subset of the HowTo100M dataset, consisting of 80 videos with 49K sentences that have been **manually annotated to rectify the alignment** in the presence of noisy correspondence. The annotators have two main tasks: i) determining if a sentence from ASR is visually related to the video, and ii) adjusting the start \& end timestamps to accurately cover the visual content if the sentence is related.
>
> After training the models on the HowTo100M dataset, we evaluated their performance on this alignment task to assess their ability to handle noise. We report the recall metrics for this alignment task. Specifically, for a **misaligned sentence**, if its most closely matched video frame falls into the ground-truth segment annotated by the human, it is counted as a successful recall. We employ a sliding window approach to calculate the similarity between video frames and sentences with a window size of 32 seconds and a step size of 8 seconds. We averaged the similarity scores for overlapping visual tokens from multiple windows.
>
> As shown below, CLIP exhibits inferior performance, possibly because it has only been trained on images and lacks the ability to capture video dynamics. In contrast, our method outperforms VideoCLIP and TempCLR, providing evidence that our approach is not prone to fit noisy correspondence.
>
> | Approach                               |  Recall  |
> | :------------------------------------- | :------: |
> | CLIP (ViT-B/32) (Radford et al., 2021) |   17.5   |
> | MIL-NCE (Miech et al., 2020)           |   34.2   |
> | TAN (Han et al., 2022)                 |   41.1   |
> | VideoCLIP (Xu et al., 2021)            |   44.4   |
> | TempCLR (Yang et al., 2023)            |   44.1   |
> | Norton (Ours)                          | **46.9** |
>
> [A] Tengda Han, Weidi Xie, and Andrew Zisserman. Temporal alignment network for long-term video. In Proceedings of the IEEE/CVF Conference on Computer Vision and Pattern Recognition, 2022.
>
>
>
> > ***Question 2:** In certain scenarios, such as video-paragraph retrieval on YouCookII (as shown in Table 1) and in the context of ablation experiments (as indicated in Table 7), we observe encouraging indications of potential improvements. While in other experiments (as depicted in Table 2), there are mixed results across various metrics. Can you shed light on the reasons behind this disparity?*
>
> Thanks for your feedback. After a thorough review, we confirm that our results for video-paragraph retrieval with background (Table 2) exhibit consistent and noteworthy performance. In the table below, we have marked the second-best results with a star (*). As shown, our method consistently achieves the best performance across various metrics, with the exception of R@10 under the DTW and OTAM metrics compared to VideoCLIP.
>
> We would like to emphasize the importance of the retrieval metric R@1, which indicates how often the correct prediction is the first result—a critical factor in many practical applications. In contrast, R@10 provides a broader perspective and may be less critical, as users typically focus on the top few results in practical scenarios. Importantly, we have surpassed VideoCLIP by **20.4% and 17%** in terms of R@1, with only a marginal drop of 0.5% and 1.2% on R@10 under DTW and OTAM, respectively.
>
> | Approach      | R@1       | R@5      | R@10     |
> | ------------- | --------- | -------- | -------- |
> |               | Cap. Avg. |          |          |
> | VideoCLIP     | 73.6*     | **94.7** | **98.4** |
> | TempCLR       | 71.7      | 94.5     | 97.9     |
> | Norton (Ours) | **74.8**  | **94.7** | **98.4** |
> |               | DTW       |          |          |
> | VideoCLIP     | 55.7      | 93.1     | **98.9** |
> | TempCLR       | 70.4*     | 93.8*    | 97.9     |
> | Norton (Ours) | **76.1**  | **95.0** | 98.4*    |
> |               | OTAM      |          |          |
> | VideoCLIP     | 56.6      | 92.8     | **98.9** |
> | TempCLR       | 72.2*     | 94.5*    | 97.7*    |
> | Norton (Ours) | **73.6**  | **94.7** | 97.7*    |

---

> ### Author Response · Authors · 2023-11-17
> **The Response to Reviewer TtfV (Part 2/2)**
>
> ##
>
> > ***Question 3:** How does the proposed method compare with other OT-based methods like action sequence matching? What are the benefits or drawbacks of using OT for video-text learning? Please provide some comparisons and discussions.*
>
> **Benefits and drawbacks of employing Optimal Transport (OT).** Our work specifically concentrates on learning temporal correspondence from long videos and addresses the challenges posed by noisy correspondence, as illustrated in Fig. 1 of the manuscript. The benefits of leveraging OT in video-text learning are evident in its inherent ability to address *asynchronous misalignment and one-to-many alignment* issues, e.g., text $\mathbf{t}_3$ is originally aligned with $\mathbf{v}_3$  but should be realigned with both $\mathbf{v}_4$ and $\mathbf{v}_5$  in Fig. 1. However, OT comes with certain drawbacks in this context,
>
> 1. Fine-Grained Alignment. OT estimates sequence distance based on clip-caption similarity, leaving the fine-grained word-frame misalignment problem unexplored.
> 2. Strict Instance Mapping. OT requires each source instance to exactly map to the targets, which is impractical when dealing with a large amount of meaningless text.
>
> To overcome these challenges, our proposed method introduces a soft-maximum operator for fine-grained alignment and an alignment prompt bucket to filter out meaningless clips and captions within the OT framework. **These unique components distinguish our method from previous OT-based methods**, enabling the effective handling of noisy correspondence and achieving superior results in video understanding.
>
> **Comparisons with other OT-based methods**. We compared our method with OT-based sequence matching methods, specifically Su & Hua (2017) [ref.B] and Liu et al. (2022) [ref.C]. Su & Hua (2017) introduce an OT method for action sequence matching with novel temporal regularization terms, aiming to encourage transports to nearby instances and penalize alignments between distant instances. Liu et al. (2022) also propose temporal priors on the optimal transport.
>
> We re-implemented Su & Hua (2017) in the context of video learning and evaluated it on the YouCookII dataset. The table below presents results for clip-caption retrieval (marked as "Clip"), video-paragraph retrieval with video backgrounds (marked as "Video (w B)"), and video-paragraph retrieval without video backgrounds (marked as "Video (w/o B)"). Notably, our proposed Norton outperforms Su & Hua (2017) by a significant margin in both short and long video retrieval tasks. As Liu et al. (2022) did not provide the training code, we compared our method with Liu et al. (2022) on the action segmentation dataset COIN based on its reported results. Specifically, we achieved a frame-wise accuracy of 69.8, while Liu et al. (2022) obtained 47.3.
>
> | Method          |   Clip   |   Clip   | Video (w/o B) | Video (w/o B) | Video  (w B) | Video  (w B) |
> | :-------------- | :------: | :------: | :-----------: | :-----------: | :----------: | :----------: |
> |                 |   R@1    |   R@5    |      R@1      |      R@5      |     R@1      |     R@5      |
> | Su & Hua (2017) |   23.3   |   50.3   |     84.4      |     97.2      |     74.3     |     94.3     |
> | Norton (Ours)   | **24.2** | **51.9** |   **88.7**    |   **98.8**    |   **76.1**   |   **95.0**   |
>
>
>
> [B] Bing Su and Gang Hua. Order-preserving wasserstein distance for sequence matching. In Proceedings of the IEEE conference on computer vision and pattern recognition, pp. 1049–1057, 2017.
>
> [C] Weizhe Liu, Bugra Tekin, Huseyin Coskun, Vibhav Vineet, Pascal Fua, and Marc Pollefeys. Learning to align sequential actions in the wild. In Proceedings of the IEEE/CVF Conference on Computer Vision and Pattern Recognition, pp. 2181–2191, 2022

---

### Official Review · Reviewer_UrFs · 2023-11-01

**Soundness:** 3 good
**Presentation:** 3 good
**Contribution:** 3 good
**Rating:** 8
**Confidence:** 3

**Summary:**

The paper introduces NOise Robust Temporal Optimal traNsport (Norton), a method designed to address multi-granularity noisy correspondence (MNC) in video-language studies, particularly focusing on long-term temporal dependencies in long videos. Norton utilizes optimal transport (OT) to handle both coarse-grained clip-caption misalignment and fine-grained frame-word misalignment, hindering temporal learning and video understanding. The method incorporates video-paragraph and clip-caption contrastive losses, an alignable prompt bucket for filtering irrelevant clips and captions, and a soft-maximum operator for identifying crucial words and keyframes. Additionally, Norton rectifies alignment targets in clip-caption contrastive learning to ensure precise temporal modeling. The effectiveness of Norton is demonstrated through extensive experiments on video retrieval, video QA, and action segmentation tasks.

**Strengths:**

1. The paper introduces a novel method, NOise Robust Temporal Optimal traNsport (Norton), which addresses the multi-granularity noisy correspondence (MNC) problem in video-language studies. This method is unique in its approach to handling both coarse-grained and fine-grained misalignments using optimal transport.
2. Norton incorporates innovative components such as an alignable prompt bucket and a soft-maximum operator to address specific challenges in video-language representation learning.
3. The method is rigorously evaluated across various tasks, including video retrieval, videoQA, and action segmentation, demonstrating its effectiveness and robustness.
4. The paper provides extensive experimental results, comparisons with existing methods, and visualizations to validate the proposed approach.
5. The paper effectively communicates the core ideas, methodologies, and results, making it accessible to readers with a background in the field.
6. Norton addresses a significant challenge in video-language studies, particularly the handling of long-term temporal dependencies in extended videos.
7. The method’s ability to improve temporal learning and video understanding has potential implications for various applications in computer vision and natural language processing.

**Weaknesses:**

1. The paper could benefit from providing more context and justification for the chosen methodologies and design decisions. For instance, the rationale behind the specific components of the Norton method, such as the alignable prompt bucket and the soft-maximum operator, could be elaborated upon to give readers a deeper understanding of their significance and contribution to the overall approach.

2. The paper could be improved by including a more thorough discussion of the limitations of the proposed method. Acknowledging and addressing potential shortcomings or challenges in the approach would provide a more balanced view and help to set realistic expectations for the method’s applicability and performance.

3. Providing more detailed implementation details, including hyperparameter settings, training procedures, and computational resources, would enhance the reproducibility of the results and allow other researchers to more easily build upon the work.

**Questions:**

1. Could you provide more details on the design choices behind the Norton method, specifically the alignable prompt bucket and the soft-maximum operator? Understanding the rationale behind these components could offer deeper insights into their roles and contributions to the overall approach.

2. The paper presents a comparison with existing methods, but could you elaborate on specific scenarios or cases where Norton particularly excels or struggles? This information would help in understanding the practical implications and limitations of the method.

3. Could you discuss any potential limitations or challenges associated with the Norton method? Acknowledging these aspects would provide a more balanced view of the method and help set realistic expectations for its performance.

4. Could you provide more detailed implementation details, including hyperparameter settings, training procedures, and computational resources? This information would enhance the reproducibility of the results and facilitate future research building upon this work.

5. Could you elaborate on the potential implications and applications of the Norton method in real-world scenarios? Understanding the practical impact of the method could offer additional motivation for the work and highlight its significance in the field.

---

> ### Author Response · Authors · 2023-11-17
> **The Response to Reviewer UrFs (Part 1/2)**
>
> Thank you for acknowledging the novelty of our method. We will answer the questions one by one.
>
> >  ***Question 1:** Could you provide more details on the design choices behind the Norton method, specifically the alignable prompt bucket and the soft-maximum operator?*
>
> This work focuses on learning temporal correspondence from long videos, particularly addressing the challenges posed by multi-granularity noisy correspondence (MNC). As illustrated in Fig. 1 of the manuscript, MNC encompasses both coarse-grained (clip-caption) misalignment and fine-grained misalignment (frame-word). The design choices of our method are closely linked to addressing this daunting challenge.
>
> - **Overall framework.** The overall design is motivated by the phenomenon of coarse-grained asynchronous misalignment. For instance, in Fig. 1, text $\mathbf{t}_3$ is originally aligned with $\mathbf{v}_3$  but it should be realigned with both $\mathbf{v}_4$ and $\mathbf{v}_5$. Leveraging optimal transport (OT) is key as it naturally handles these **asynchronous and one-to-many alignment** challenges. To further address fine-grained misalignment and coarse-grained irrelevant misalignment, we integrate the soft-maximum operator and alignable prompt bucket into the OT framework.
> - **Soft-maximum operator.** This component is designed to identify crucial words and key frames in fine-grained frame-word alignment. As depicted in Fig. 1, only certain words in a caption are relevant to the associated video clip. Therefore, we employ the **log-sum-exp approximation** as the soft-maximum operator to identify the most significant words for each frame based on fine-grained frame-word interaction. This operator computes the clip-caption similarity for subsequent coarse-grained sequence alignment, unifying multi-grained correspondence learning within the OT framework.
> - **Alignable prompt bucket.** This component is designed to filter out irrelevant clips and captions during coarse-grained clip-caption alignment. The prompt bucket serves as a candidate alignable target for irrelevant clips and captions. The value $p$ of the bucket functions as the **similarity margin** that distinguishes between alignable and unalignable clips and captions. If a video clip lacks an alignable caption, its pairwise similarities with the set of captions are typically small. During Sinkhorn iterations, if the margin $p$ significantly exceeds these pairwise similarity values, the video clip aligns with the prompt bucket. By discarding clips/captions aligned to the prompt bucket, Norton effectively filters out noisy clips and captions, enhancing the overall alignment results.
>
>
>
> > ***Questions 2 and 5:** Could you elaborate on the potential implications and applications of the Norton method in real-world scenarios? Could you elaborate on specific scenarios or cases where Norton particularly excels or struggles?*
>
> Thanks for the suggestions. We have included a section on the potential implications and applications of Norton in real-world scenarios in Appendix E for your reference. Below is the newly added content.
>
> **Application scenarios.** Norton is a representation learning method that exhibits versatility across various tasks including video retrieval, video QA, and classification, as confirmed by our experiments. A notable strength of Norton lies in its ability to effectively address the common challenge of noisy correspondence, particularly in **uncurated instructional videos**. This adaptability allows Norton to be implemented in diverse scenarios without necessitating meticulous video curation. For instance, Norton proves effective in tasks such as long video retrieval or classification for various content genres like movies, education videos, and cooking tutorials. It's also essential to acknowledge that Norton is tailored for representation learning and may exhibit suboptimal performance in tasks focused on content generation, such as video captioning.
>
> **Potential implications.** This paper delves into two challenging problems in video understanding, namely, long video learning and noisy correspondence learning.  In addressing the former, where computational constraints have limited prior works, our proposed efficient solution may spark increased interest in long video understanding tasks. Regarding the latter, the **noisy correspondence (mismatched data pairs)** problem has garnered attention in diverse multi-modal applications, extending **beyond video-text domains** to encompass challenges in image-text retrieval, cross-modal generation, and person re-identification. Our work has the potential to attract increased attention to the broader spectrum of noisy correspondence challenges across various domains.

---

> ### Author Response · Authors · 2023-11-17
> **The Response to Reviewer UrFs (Part 2/2)**
>
> > ***Question 3:**  Could you discuss any potential limitations or challenges associated with the Norton method?*
>
> We appreciate the insightful question regarding potential challenges associated with the Norton method. The following points address these aspects and are included in Appendix F.
>
> 1. **Multi-modal scenarios**. Our approach introduces an optimal transport solution to address the noisy correspondence between bi-modalities in videos and text. However, as videos inherently encompass visual, textual, and audio content, the noisy correspondence challenge might extend across multiple modalities. Addressing multi-modal noisy correspondence using optimal transport presents an open challenge, given the quadratic growth in combinations concerning the number of modalities. We acknowledge this limitation and plan to extend our method to effectively tackle multi-modal noisy correspondence, exploring these scenarios in future work.
> 2. **Utilization of Noise.** In this paper, we employ the prompt bucket to directly filter out irrelevant clips and captions during sequential alignment, attempting to mitigate the influence of noisy correspondence. However, an intriguing question arises regarding whether these noisy samples could be utilized as an incentive for training. Exploring the possibility of generating associated text for unalignable video clips using large multimodal models (LMMs), e.g., LLaVA and GPT-4V (vision), could open up a novel avenue for exploration and improvement in future research endeavors.
>
>
>
> > ***Question 4:**  Could you provide more detailed implementation details, including hyperparameter settings, training procedures, and computational resources?*
>
> Comprehensive implementation details including hyperparameter settings, training procedures, and computational resources are available in Appendix A. Additionally, detailed training costs for several variants of our method are provided in Appendix C. For reproducibility, we commit to releasing the complete code on GitHub upon acceptance.
>
> ######

---

> > ### Comment · Reviewer_UrFs · 2023-11-20
> >
> > The authors have addressed most of my concerns. Considering their responses and the feedback from other reviewers, I am revising my final score upwards.

---

> > > ### Author Response · Authors · 2023-11-20
> > >
> > > We sincerely thank you for your positive recognition and evaluation of our work!

---

### Meta-Review · Area_Chair_ZgAR · 2023-12-03

**Metareview:**

The reviewers reach a consensus that the studied problem is significant, the proposed method is interesting and effective, and the presentation quality is high. According to my own reading, I agree with the reviewers that this is a nice paper with solid contribution to Video-language pre-training. Thus, I would sincerely recommend accepting the paper.

**Justification For Why Not Higher Score:**

N/A

**Justification For Why Not Lower Score:**

The reviewers reach a consensus that the studied problem is significant, the proposed method is interesting and effective, and the presentation quality is high. According to my own reading, I agree with the reviewers that this is a nice paper with solid contribution to Video-language pre-training.

---

### Decision · Program_Chairs · 2024-01-16

Accept (oral)